# PREDICTING LLM OUTPUT LENGTH VIA ENTROPY-GUIDED REPRESENTATIONS

**Huanyi Xie**[1,2]**, Yubin Chen**[1,2]**, Liangyu Wang**[1,2]**, Lijie Hu**[3]**, and Di Wang**[1,2,†]

[1]King Abdullah University of Science and Technology (KAUST)
[2]Provable Responsible AI and Data Analytics (PRADA) Lab
[3]Mohamed bin Zayed University of Artificial Intelligence (MBZUAI)

## ABSTRACT

The long-tailed distribution of sequence lengths in LLM serving and reinforcement learning (RL) sampling causes significant computational waste due to excessive padding in batched inference. Existing methods rely on auxiliary models for static length prediction, but they incur high overhead, generalize poorly, and fail in stochastic "one-to-many" sampling scenarios. We introduce a lightweight framework that reuses the main model's internal hidden states for efficient length prediction. Our framework features two core components: 1) Entropy-Guided Token Pooling (EGTP), which uses on-the-fly activations and token entropy for highly accurate static prediction with negligible cost, and 2) Progressive Length Prediction (PLP), which dynamically estimates the remaining length at each decoding step to handle stochastic generation. To validate our approach, we build and release ForeLen, a comprehensive benchmark with long-sequence, Chain-of-Thought, and RL data. On ForeLen, EGTP achieves state-of-the-art accuracy, reducing MAE by 29.16% over the best baseline. Integrating our methods with a length-aware scheduler yields significant end-to-end throughput gains. Our work provides a new technical and evaluation baseline for efficient LLM inference.

## 1 INTRODUCTION

In recent years, large language models (LLMs) (Achiam et al., 2023; Brown et al., 2020) have rapidly proliferated across diverse applications including chatbots (Yang et al., 2025a), code assistants (Petrovic et al., 2025), retrieval-augmented generation (RAG) (Li et al., 2025a) applications, and intelligent agents (Schmidgall et al., 2025). Supporting these applications is an efficient LLM serving infrastructure, and the typical LLM serving process follows an autoregressive paradigm: the system receives prompts generated by users or tasks, and the model constructs complete responses through iterative next-token prediction (Zhen et al., 2025; Kwon et al., 2023; Liu et al., 2025). Concurrently, LLM inference capabilities are being integrated into online reinforcement learning, such as Group Relative Policy Optimization (GRPO) (Shao et al., 2024) and its variants (Yu et al., 2025; Zheng et al., 2025). To construct stable and diverse reward signals, systems must perform multiple independent sampling operations on the same prompt, generating a set of candidate responses and computing rewards based on their relative quality, which are then fed back to the policy update phases.

In real scenarios, batching techniques (Dong et al., 2025; Xuan et al., 2025) are the core mechanism for boosting hardware utilization and overall throughput. By executing multiple requests in parallel, systems can significantly amortize scheduling and memory-access overhead. However, the generation lengths of different requests within a batch usually vary greatly (as shown in Figure 1a). Because tensor shapes must align, shorter sequences are padded to match the longest one (Gururangan et al., 2020; Yun et al., 2024). This leads to a marked "barrel effect". Redundant padding computations enlarge GPU or accelerator time consumption (Deshmukh et al., 2025), and also reduce effective computation (Qiu et al., 2024; Piotrowski et al., 2025). If systems could estimate output lengths for each request before or early in inference, they could apply length-aware scheduling. Such scheduling would cut ineffective computation. It would also raise throughput and cost

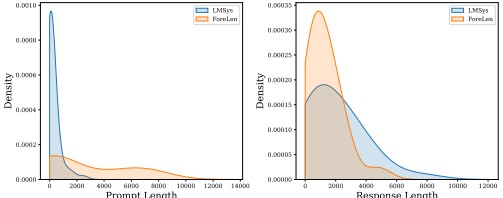
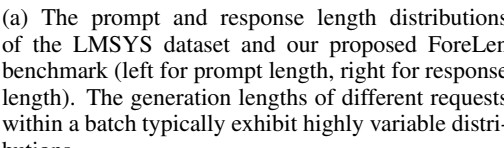
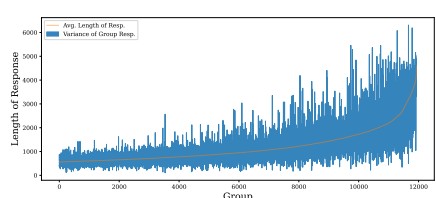

(a) The prompt and response length distributions of the LMSYS dataset and our proposed ForeLen benchmark (left for prompt length, right for response length). The generation lengths of different requests within a batch typically exhibit highly variable distributions.

(b) Response length distribution observed during a GRPO training loop for online RL sampling. Despite using identical prompts within each group, stochastic decoding introduces substantial variability in generated response lengths, highlighting the challenge for dynamic length prediction.

Figure 1: **Analysis of LLM Sequence Length Distributions in Representative Scenarios.** Figure (a) illustrates the length characteristics of our proposed **ForeLen** benchmark and LMSYS, demonstrating a wider and longer-tailed distribution. Figure (b) showcases the high length variance for generations from identical prompts in an online RL training setting.

efficiency in both online serving (Jin et al., 2023) and RL training (Zheng et al., 2025; Yu et al., 2025).

To predict output lengths for LLMs, recent studies (Qiu et al., 2024; Jin et al., 2023; Hu et al., 2024; Fu et al., 2024) typically attach a fine-tuned, lightweight auxiliary predictor, e.g., DistilBERT (Sanh et al., 2019) or OPT (Zhang et al., 2022). Although this design is appealing, it still suffers from three key limitations: (i) Instability in stochastic, "one-to-many" generation. During sampling—especially in reinforcement-learning workflows (Wang et al., 2025)—a single prompt can yield multiple valid completions with widely differing lengths (Figure 1b). Any static estimate that relies only on the prompt therefore becomes unreliable. (ii) Limited accuracy and generalization. These predictors are usually trained on benchmarks such as LMSYS (Zheng et al., 2024), which contain few long sequences and little complex reasoning. As a result, their length forecasts deteriorate in realistic, more complex settings. (iii) Additional computational and deployment cost. Each request must run a separate predictor instead of reusing the rich hidden states already produced by the main LLM.

To overcome these challenges of inefficiency, generalization, and inflexibility, we propose a new framework that directly utilizes the information embedded within the LLM's internal activations. Our core insight is that if an LLM can determine when to emit the `<eos>` token, then signals related to the eventual output length must be implicitly encoded in its internal states. By reusing these activations, we can enable more accurate length prediction with minimal additional cost. We introduce Entropy-Guided Token Pooling (EGTP), which reuses on-the-fly activations, guided by token entropy, to capture the most informative signals from the prompt, thereby addressing the overhead and generalization challenges of prior work.

Furthermore, to address the fundamental difficulty of length prediction in stochastic environments, we introduce Progressive Length Prediction (PLP). PLP leverages the autoregressive nature of LLMs by operating dynamically at each decoding step. It uses the current activations to produce an updated estimate of the remaining tokens to be generated. By iteratively refining its forecast, PLP enables length-aware scheduling even in highly unpredictable environments like RL sampling, a task for which previous static methods are not inherently designed.

To rigorously validate our framework, particularly in scenarios where existing methods may struggle, we introduce ForeLen, the first comprehensive benchmark for length prediction featuring long-sequence, Chain-of-Thought (CoT), and reinforcement learning (RL) sampling data. Experiments on ForeLen show that our method, EGTP, achieves state-of-the-art accuracy. Averaged across all tested models, our method reduces the Mean Absolute Error (MAE) by 29.16% compared to the strongest baseline and 55.09% compared to the widely-used SSJF-Reg (Qiu et al., 2024). This superior prediction accuracy, in turn, yields significant improvements in end-to-end inference throughput.

Our contributions are summarized as follows:

- We propose a lightweight and efficient length prediction framework, comprising two core modules: **EGTP**, which reuses internal model activations guided by token entropy for accurate static prediction, and **PLP**, which performs progressive prediction to handle highly stochastic RL sampling.

- To facilitate rigorous evaluation, we construct and release **ForeLen**, the first comprehensive benchmark designed to test predictors on challenging long-sequence, CoT, and RL data.

- Our methods achieve state-of-the-art prediction accuracy on ForeLen and, when integrated with a length-aware scheduler, yield significant improvements in inference throughput.

## 2 RELATED WORK

**Efficient LLM Serving and Inference Optimization.** Efficient LLM serving relies on optimizations like continuous batching (e.g., in vLLM) (Kwon et al., 2023) and efficient KV Cache management like PagedAttention (Kwon et al., 2023). These techniques maximize throughput by dynamically managing requests and mitigating memory fragmentation. However, while they reduce inter-request idle time, they do not solve the computational waste from padding within a running batch, known as the "barrel effect," where shorter sequences waste computation matching the longest (Deshmukh et al., 2025). Our work is orthogonal: by predicting output length, we enable length-aware schedulers to build more homogeneous batches, directly reducing this padding overhead and complementing existing serving architectures.

**LLM Response Length Prediction.** Prior work on length prediction primarily trains lightweight auxiliary models (e.g., DistilBERT) to predict an LLM's output length based only on the input prompt (Qiu et al., 2024; Jin et al., 2023; Hu et al., 2024). This approach, however, incurs non-negligible overhead, requiring a separate model to be trained, deployed, and executed for every request. We bypass this cost. Inspired by work linking internal model states like token entropy to generation structure (Li et al., 2025c), our EGTP (Entropy-Guided token Pooling) method reuses the LLM's own on-the-fly activations to achieve accurate prediction with negligible additional computation.

**LLM Inference and Sampling in Reinforcement Learning.** Modern LLM alignment algorithms like GRPO (Shao et al., 2024) and its variants (Yu et al., 2025; Zheng et al., 2025) require generating multiple candidate responses from the same prompt using stochastic sampling. This process creates extreme output length variance for a single input, rendering all existing static, prompt-based predictors (Qiu et al., 2024) completely ineffective. Furthermore, a standardized benchmark for this "one-to-many" prediction scenario is absent (Wang et al., 2025). Our PLP (Progressive Length Prediction) is designed specifically for this dynamic setting. Instead of a single static forecast, it operates autoregressively, using current model activations at each step to iteratively update its prediction of the remaining tokens, thereby adapting to the unique path of each stochastic sample.

## 3 METHOD

Our proposed methodology tackles the challenge of length prediction through two primary components designed for static and dynamic scenarios, respectively. First, for static prediction from complex prompts, we introduce Entropy-Guided Token Pooling (EGTP). This method efficiently utilizes the LLM's internal activations, and its accuracy is further boosted by our novel Regression via Soft Label Distribution training strategy. Second, to handle the 'one-to-many' problem in stochastic environments like RL, we present Progressive Length Prediction (PLP), a dynamic approach that iteratively refines its forecast. This capability is crucial for scenarios unaddressed by prior static methods.

### 3.1 ENTROPY-GUIDED TOKEN POOLING (EGTP)

**Motivation.** As mentioned above, our core insight is that if an LLM can determine when to emit the `<eos>` token, then signals related to the eventual output length must be implicitly encoded in its internal states. By reusing these activations, we can enable accurate length prediction with minimal additional cost. However, a challenge arises because we have a sequence of these representations,

but we need to produce a single predictive output. Therefore, we require a pooling mechanism to aggregate them into one conclusive vector. We find that traditional methods like mean or max pooling are suboptimal, as they can dilute or discard crucial information. Instead, we propose a novel pooling strategy guided by predictive entropy, which we believe more effectively captures the most informative tokens to create a superior final representation.

To validate this hypothesis, we measured each token's contribution to the final prediction using a **gradient-based attribution** method (Sundararajan et al., 2017). Specifically, we define the importance $I_t$ of a token at position $t$ as the L2 norm of the gradient of the Mean Squared Error (MSE) loss ($\mathcal{L}_{\text{MSE}}$) between the predicted and ground truth lengths, with respect to its hidden state representation $h_t$:

$$I_t = \|\nabla_{h_t} \mathcal{L}_{\text{MSE}}\|_2. \tag{1}$$

To provide an empirical foundation for our approach, we analyzed the relationship between token entropy and importance. The experiment was performed using a BERT model with a linear head on 10k samples randomly selected from the LMSYS dataset. As shown in Figure 2, the results demonstrate a significant positive correlation between a token's entropy and its importance for length prediction (Pearson's $r = 0.451$). This finding confirms that high-entropy tokens, which are those where the model is most uncertain about what to generate next, are indeed critical signals for forecasting the output length, thus validating our entropy-weighted pooling strategy.

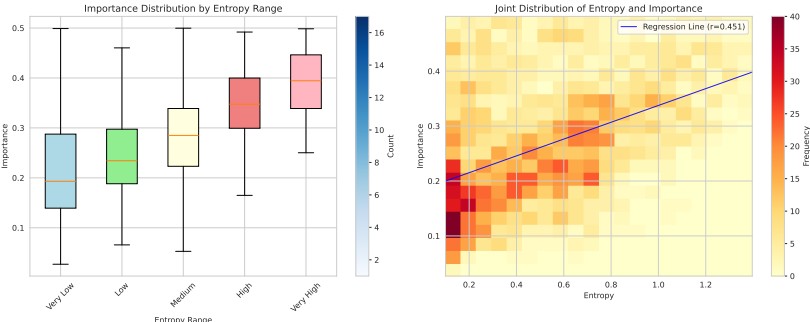

Figure 2: **Empirical validation of the relationship between token entropy and importance. (Left)** The box chart shows the average importance for tokens binned into five equal-width intervals based on their entropy value. This indicates that importance increases with entropy. **(Right)** The scatter plot displays the joint distribution of entropy and importance, with the regression line confirming a significant positive correlation (r=0.451).

**EGTP.** Based on this validation, we propose **Entropy-Guided Token Pooling (EGTP)**. This method aggregates a sequence of input hidden states $\{h_1, h_2, \ldots, h_n\}$ into a single feature vector $\mathbf{h}$. The process begins by computing the entropy $H_i$ for each token. Specifically, for each hidden state $h_i$ corresponding to an input token $x_i$, we calculate the entropy from the next-token probability distribution $P(v|x_{<i})$ over the vocabulary $V$:

$$H_i = -\sum_{v \in V} P(v|x_{<i}) \log P(v|x_{<i}) \tag{2}$$

Next, these entropy values are used to generate attention weights. We transform the entropies into a distribution of weights $w_i$ via a softmax function, scaled by a temperature parameter $\alpha$ that controls the distribution's sharpness:

$$w_i = \frac{\exp(H_i)}{\sum_{j=1}^n \exp(H_j)} \tag{3}$$

Finally, the aggregated representation $\mathbf{h}$ is computed as the weighted sum of the hidden states, using the entropy-derived weights:

$$\mathbf{h} = \sum_{i=1}^n w_i h_i \tag{4}$$

By adaptively focusing on the most informative parts of the input prompt, EGTP provides a higher-quality feature representation for the downstream length prediction task.

## 3.2 Length Regression via Soft Label Distribution

Next, we will use our above feature representations to build a model for length prediction. While length prediction is a regression task, standard MSE loss is highly sensitive to outliers or heavy-tailed distributions, which have been shown in Figure 1a. The common alternative, classification via length binning, ignores the crucial concept of distance between true and predicted values. Our approach overcomes these limitations by designing a prediction head that is both robust to outliers like classification and distance-aware like regression.

Our method begins by converting the continuous length target $y$ into a soft probability distribution $\mathbf{p}$ to serve as the ground truth label. We first discretize the target space into $K$ predefined bins. Instead of using a one-hot vector, we generate a soft distribution where, for a true length falling into bin $i$, the probability $p_j$ for any bin $j$ is inversely proportional to its distance from bin $i$. This is formulated as:

$$p_j = \frac{\exp(-|j - i|)}{\sum_{k=1}^{K} \exp(-|k - i|)} \tag{5}$$

Next, using the feature vector $\mathbf{h}$ from EGTP, our model produces two concurrent outputs. The first is a **classification prediction** $\hat{\mathbf{p}}$, which is a $K$-dimensional probability distribution $[\hat{p}_1, \ldots, \hat{p}_K]$ obtained via a softmax layer. The second is the final **regression prediction** $\hat{y}$, which is calculated as the expected value of the predicted distribution. Assuming $c_i$ is the center value of the $i$-th bin, this is computed as:

$$\hat{y} = \sum_{i=1}^{K} \hat{p}_i \cdot c_i \tag{6}$$

Finally, the model is trained by optimizing a joint loss function that combines a Cross-Entropy (CE) loss and a Mean Squared Error (MSE) loss, balanced by a hyperparameter $\lambda$:

$$\mathcal{L} = \lambda \mathcal{L}_{\text{CE}}(\mathbf{p}, \hat{\mathbf{p}}) + (1 - \lambda)\mathcal{L}_{\text{MSE}}(y, \hat{y}) \tag{7}$$

The $\mathcal{L}_{\text{CE}}$ term encourages the predicted distribution $\hat{\mathbf{p}}$ to align with the soft label distribution $\mathbf{p}$, thereby providing stable, gradient-friendly supervision. Simultaneously, the $\mathcal{L}_{\text{MSE}}$ term directly minimizes the error between the final continuous prediction $\hat{y}$ and the true length $y$, ensuring regression accuracy.

## 3.3 Progressive Length Prediction (PLP)

In scenarios such as online reinforcement learning, a system often generates multiple candidate responses with varying lengths from a single prompt. In this context, a static, pre-generation prediction is insufficient. To address this, we introduce **Progressive Length Prediction (PLP)**. PLP leverages the autoregressive nature of LLMs by making a new prediction at **each decoding step**. At timestep $t$, its objective is to predict the *remaining* number of tokens to be generated $y_{\text{rem}}^{(t)}$. This is done to leverage the information from all previously generated tokens to make a more accurate prediction at each step.

To do this, PLP first forms a dynamic input representation $z_t$ by combining the prompt feature vector $\mathbf{h}$ with the hidden states of the already generated tokens $\{h'_1, \ldots, h'_t\}$:

$$z_t = \text{Aggregate}(\mathbf{h}, \{h'_1, \ldots, h'_t\}) \tag{8}$$

where $\text{Aggregate}(\cdot)$ is a simple concatenation function. This representation $z_t$ is then passed through the same prediction head described in Section 3.2 to yield the final prediction for the remaining length, $\hat{y}_{\text{rem}}^{(t)}$.

The model is trained by minimizing the average loss across all timesteps. The total loss for a single sequence is:

$$\mathcal{L}_{\text{PLP}} = \frac{1}{T} \sum_{t=1}^{T} \mathcal{L}(y_{\text{rem}}^{(t)}, \hat{y}_{\text{rem}}^{(t)}) \tag{9}$$

where $T$ is the total sequence length and $\mathcal{L}$ is the joint loss function defined in Eq. (7). By iteratively refining its forecast, PLP enables dynamic adjustments to scheduling strategies, thereby improving resource utilization.

# 4 EXPERIMENTS

## 4.1 DATASET CONSTRUCTION

To evaluate our proposed method in complex settings, we constructed the **ForeLen**, which comprises two core scenarios designed to comprehensively assess predictor performance under challenging conditions.

**Scenario 1: Long-Sequence and Complex Reasoning Generation.** This scenario focuses on long-text and complex reasoning capabilities. We selected prompts from LongBench (Bai et al., 2024a), ZeroSCROLLS (Shaham et al., 2023), and IFEval (Zhou et al., 2023). For long-sequence tasks, we used the Qwen2.5 (0.5B-7B) (Yang et al., 2025b) and Llama3.2 (1B, 3B) (Dubey et al., 2024) model series to generate outputs. For reasoning tasks, outputs were generated by the Qwen2.5 and DeepSeek-R1-Distill model series (Guo et al., 2025).

**Scenario 2: Dynamic RL Sampling.** This scenario is designed to simulate the dynamic sampling process in RL training. We collected data from the actual GRPO training pipeline of the Qwen2.5 and Llama3.2 model series. Prompts were sourced from six widely-used math and code reasoning datasets: CRUXEval (Gu et al., 2024), GSM8K (Cobbe et al., 2021), Live-CodeBench (Jain et al., 2025), MATH (Hendrycks et al., 2021b), MBPP (Austin et al., 2021), and MMLU-STEM (Hendrycks et al., 2021a). For each prompt, we applied a grouped sampling strategy with K=4 and recorded generated candidate responses and lengths.

**Data Splits and Statistics.** To ensure a fair evaluation, we strictly adhere to the official train/val/test splits of the source datasets. This guarantees that prompts in the validation and test sets are unseen during the predictor's training phase. Detailed statistics of the dataset are presented in Appendix Table 4.

## 4.2 EXPERIMENT SETTING

**Baselines and Additional Datasets.** We compare our method against baselines including SSJF-Reg (Qiu et al., 2024), SSJF-MC (Qiu et al., 2024), S3 (Jin et al., 2023), PiA (Zheng et al., 2023), TPV (Eisenstadt et al., 2025), TRAIL (Shahout et al., 2025), and LTR-C (Fu et al., 2024). Details about these baselines are shown in Appendix E.1. Our evaluation is conducted on the popular LMSYS (Zheng et al., 2024) benchmark, as well as our ForeLen benchmark, which is designed to be richer and more challenging.

**Evaluation Metrics.** We use the Mean Absolute Error (MAE) to evaluate the performance of our method. Use throughput, Job Completion Time and padding ratio to evaluate the end-to-end system performance. Details are shown in Appendix B.

**Experimental Setup.** For training, we use the AdamW (Kingma & Ba, 2015; Loshchilov & Hutter, 2019) optimizer with a learning rate of **2e-5**. We train the model for a maximum of 10 epochs with a batch size of 16. For reproducibility across all experiments, we set the random seed to 42. For the Soft Label Regression specific settings, the target length is discretized into $K = 20$ bins. And we set $\lambda$ to 0.95 to balance the CE loss and MSE loss. Experiments are run with 1 V100 GPU, 10 core CPU, and 64 GB memory. For all baseline methods, we adopt the hyperparameter settings reported in their original papers.

## 4.3 MAIN RESULTS: PREDICTION ACCURACY

To evaluate the efficacy of our proposed method, EGTP, we conducted a comprehensive comparison against a suite of state-of-the-art baselines for output length prediction. As presented in Table 1, our evaluation measures the MAE on two distinct benchmarks: the widely-used LMSYS dataset and our more challenging ForeLen benchmark. The results unequivocally demonstrate that EGTP consistently and significantly outperforms all other methods across every model and scenario. On the standard LMSYS benchmark, EGTP achieves the lowest MAE when predicting output lengths for both GPT-4 (87.32) and Claude-2 (68.33), surpassing the next-best methods by 9.1% and 11.3%, re-

| Model | Scenario | Prediction Method | | | | | | | |
|---|---|---|---|---|---|---|---|---|---|
| | | EGTP(Ours) | SSJF-Reg | SSJF-MC | S3 | PiA | TPV | TRAIL | LTR-C |
| *LMSYS Benchmark* | | | | | | | | | |
| **GPT-4** | LMSYS | **87.32** | 171.62 | 190.93 | 96.03 | 143.02 | 339.88 | 116.91 | 104.11 |
| **Claude-2** | LMSYS | **68.33** | 152.00 | 140.18 | 83.51 | 91.32 | 283.81 | 102.39 | 77.03 |
| *ForeLen Benchmark* | | | | | | | | | |
| **Qwen2.5 3B** | LongSeq | **93.43** | 271.15 | 771.96 | 186.82 | 346.36 | 576.08 | 147.92 | 124.23 |
| | Reasoning | **139.04** | 325.38 | 789.21 | 169.72 | 428.10 | 621.72 | 132.20 | 145.53 |
| | RL | **99.78** | 194.03 | 187.04 | 169.306 | 197.87 | 238.20 | 159.15 | 192.84 |
| | **Avg.** | **110.75** | 263.52 | 582.74 | 175.28 | 324.11 | 478.67 | 146.42 | 154.20 |
| **Qwen2.5 7B** | Long Seq | **81.60** | 210.68 | 507.34 | 161.82 | 279.55 | 533.92 | 134.18 | 129.37 |
| | Reasoning | **133.57** | 298.80 | 770.92 | 168.80 | 412.84 | 466.56 | 124.19 | 134.55 |
| | RL | **95.24** | 177.07 | 167.18 | 173.37 | 212.78 | 202.98 | 155.51 | 187.13 |
| | **Avg.** | **103.47** | 228.85 | 481.81 | 168.00 | 301.72 | 401.15 | 137.96 | 150.35 |
| **Llama3.2 1B** | Long Seq | **81.77** | 173.97 | 251.71 | 262.27 | 145.61 | 512.16 | 145.35 | 179.11 |
| | Reasoning | **138.04** | 157.67 | 143.78 | 152.29 | 254.57 | 273.04 | 148.28 | 142.28 |
| | RL | **95.44** | 204.87 | 267.08 | 235.07 | 197.87 | 308.09 | 161.78 | 206.70 |
| | **Avg.** | **105.08** | 178.84 | 220.86 | 216.54 | 199.35 | 364.43 | 151.80 | 176.03 |
| **Llama3.2 3B** | Long Seq | **78.83** | 193.34 | 234.83 | 259.55 | 149.92 | 733.75 | 143.62 | 317.89 |
| | Reasoning | **111.23** | 186.16 | 160.36 | 164.22 | 264.66 | 373.03 | 177.16 | 174.01 |
| | RL | **114.53** | 418.09 | 218.00 | 163.86 | 197.87 | 244.99 | 152.85 | 131.75 |
| | **Avg.** | **101.53** | 265.86 | 204.40 | 195.88 | 204.15 | 450.59 | 157.88 | 207.88 |

Table 1: Comparison of different length prediction methods. (**Best**, **SecondBest**)

spectively. This initial result validates the fundamental effectiveness of our approach on established, real-world conversational data.

We further assessed our method on the more demanding ForeLen benchmark, which incorporates complex scenarios involving Long Sequences, Reasoning, and data from LLMs Reinforcement Learning. Even in these challenging conditions, EGTP maintains its exceptional performance and reaffirms its superiority. When analyzing the average performance across these scenarios, EGTP establishes a substantial margin over the strongest baseline, TRAIL, reducing the MAE from 146.42 to 110.75 (a 24.4% improvement) for Qwen2.5 3B; from 137.96 to 103.47 (a 25.0% improvement) for Qwen2.5 7B; from 151.80 to 105.08 (a 30.8% improvement) for Llama3.2 1B; and from 157.88 to 101.53 (a remarkable 35.7% improvement) for Llama3.2 3B. This consistent outperformance across diverse models and complex tasks highlights the excellent generalization capabilities of EGTP and strongly underscores the critical role of token entropy in providing a robust signal for accurate output length prediction.

**The Effect of PLP.** The experimental results in Figure 3 for Progressive Length Prediction show consistent improvements across all three tasks. The RL task demonstrates the strongest performance gains, dropping from 95.24 to 80.85 MAE, while both Reasoning and Long Seq tasks also show substantial improvements. All tasks exhibit a similar convergence pattern where rapid initial gains in the first few steps gradually stabilize, suggesting that PLP effectively leverages already-generated tokens to refine length predictions. The improvement across diverse task types validates PLP's core approach of progressive refinement during the decoding process, making it particularly valuable for dynamic resource allocation scenarios where accurate length prediction is crucial for efficient scheduling.

## 4.4 END-TO-END SYSTEM PERFORMANCE

To evaluate the practical effectiveness of our proposed EGTP, we integrated it and baseline predictors with a Shortest Job First (SJF) scheduler in an end-to-end system powered by the vLLM serving backend. We tested the system on two distinct workloads: Long Sequence and Reasoning. The results, presented in the Table 2, unequivocally demonstrate that EGTP consistently and substantially outperforms all baselines across every metric in both scenarios. In the Long Sequence workload, EGTP not only achieves the highest throughput but also more than halves the JCT compared to the best-performing baseline, TRAIL. This significant performance gain is directly driven by EGTP's superior prediction accuracy, which slashes the padding ratio to just 0.18, a nearly 3x reduction over

TRAIL's 0.51. This trend extends to the Reasoning scenario, where EGTP again leads all methods by reducing the padding ratio to a mere 0.09. This represents a 36% relative reduction in wasted computation over the strongest baseline, LTR-C, cementing its lead in throughput and JCT.

| Model | Long Sequence | | | Reasoning | | |
|---|---|---|---|---|---|---|
| | Throughput ↑ | Avg. JCT ↓ | Padding Ratio ↓ | Throughput ↑ | Avg. JCT ↓ | Padding Ratio ↓ |
| **EGTP (Ours)** | **131.05** | **4.20** | **0.18** | **194.21** | **8.21** | **0.09** |
| SSJF-Reg | 117.87 | 11.52 | 0.57 | 146.74 | 9.32 | 0.33 |
| SSJF-MC | 109.50 | 23.24 | 0.38 | 124.34 | 22.31 | 0.31 |
| S3 | 115.09 | 15.74 | 0.55 | 139.89 | 10.31 | 0.42 |
| PiA | 119.20 | 18.40 | 0.47 | 141.83 | 16.74 | 0.31 |
| TPV | 116.03 | 43.10 | 0.57 | 142.41 | 40.12 | 0.32 |
| TRAIL | **129.58** | **9.45** | 0.51 | 147.03 | 9.32 | 0.34 |
| LTR-C | 127.14 | 12.01 | **0.21** | **150.57** | **9.30** | **0.14** |

Table 2: End-to-End System Performance Comparison on Different Scenarios (**Best**, **Second Best**)

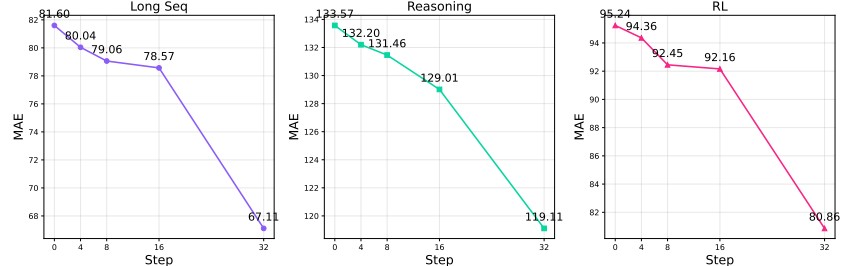

Figure 3: MAE improvements for Progressive Length Prediction.

## 4.5 ABLATION STUDY

**Effectiveness of Entropy-Guided Pooling.** Our ablation study on different pooling methods, with

Table 3: Ablation study on the impact of different pooling methods.

| Pooling Method | Reasoning | Long Sequence | RL | Average |
|---|---|---|---|---|
| **EGTP (Ours)** | **133.57** | **81.60** | **95.24** | **103.47** |
| Average Pooling | 142.40 | 173.85 | 149.92 | 155.39 |
| Max Pooling | 137.88 | 122.74 | 98.46 | 119.69 |
| Last Token Pooling | 139.09 | 135.44 | 105.39 | 126.64 |

results in Table 3, clearly demonstrates that our proposed EGTP method outperforms the baselines across all tasks. EGTP achieves an average MAE of 103.47, which is a significant improvement over the best-performing baseline, Max Pooling, at 119.69. The advantage of EGTP is especially prominent on the Long Sequence task, where its MAE of 81.60 is substantially lower than any competing method. This result confirms the superiority of our approach in effectively capturing key features from complex sequences, a task where traditional pooling strategies tend to fall short.

The sensitivity analysis of the hyperparameter $\lambda$ and its effect on the joint optimization is discussed in detail in Appendix D. Additionally, detailed experimental results for the Qwen2.5-0.5B and Qwen2.5-1.5B models can be found in Appendix E.4. And the length prediction performance comparison on various other datasets is provided in the Appendix E.5.

## 5 CONCLUSION

In this paper, we propose a novel framework that predicts sequence length by reusing the model's own internal activations. This approach circumvents the overhead and generalization failures of

separate, auxiliary predictors. Our method introduces EGTP for static estimation and PLP for progressive prediction in dynamic environments. We validate our approach on ForeLen, a new and challenging benchmark we developed for this task. The results demonstrate superior prediction accuracy, confirming that sufficient signals for length determination are already encoded within the LLM's hidden states.

## 6 ACKNOWLEDGMENT

Di Wang, Huanyi Xie, and Liangyu Wang are supported in part by the funding BAS/1/1689-01-01,RGC/3/7125-01-01, FCC/1/5940-20-05, FCC/1/5940-06-02, and King Abdullah University of Science and Technology (KAUST) – Center of Excellence for Generative AI, under award number 5940 and a gift from Google. Lijie Hu is supported by the funding BF0100 from Mohamed bin Zayed University of Artificial Intelligence (MBZUAI). For computer time, this research used lbex managed by the Supercomputing Core Laboratory at King Abdullah University of Science & Technology (KAUST) in Thuwal, Saudi Arabia.

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

# Appendix

## A  THE USE OF LARGE LANGUAGE MODELS(LLMS)

During the preparation of this manuscript, we utilized Large Language Models (LLMs) for assistance with grammar checking and text polishing to enhance the clarity and readability of the paper.

## B  EVALUATION METRICS

This section details the metrics used to evaluate the performance of our models and the end-to-end system.

### B.1  PREDICTOR PERFORMANCE METRIC

To evaluate the performance of our predictor, we use the **Mean Absolute Error (MAE)**.

- **Mean Absolute Error (MAE)** measures the average absolute difference between the predicted values ($\hat{y}_i$) and the actual ground truth values ($y_i$). A lower MAE indicates a more accurate prediction model. Given $n$ samples, it is defined as:

$$\text{MAE} = \frac{1}{n} \sum_{i=1}^{n} |y_i - \hat{y}_i|$$

### B.2  END-TO-END SYSTEM PERFORMANCE METRICS

To evaluate the overall system performance, we use throughput, Job Completion Time, and padding ratio.

- **Throughput** measures the number of jobs the system can successfully process per unit of time. It is a key indicator of system efficiency. Let $N_{\text{completed}}$ be the total number of completed jobs in a time interval $T_{\text{total}}$:

$$\text{Throughput} = \frac{N_{\text{completed}}}{T_{\text{total}}}$$

- **Job Completion Time (JCT)** refers to the total time elapsed from a job's submission ($T_{\text{submission}}$) to its completion ($T_{\text{completion}}$). We typically measure the average JCT across all jobs to gauge system responsiveness.

$$\text{JCT} = T_{\text{completion}} - T_{\text{submission}}$$

- **Padding Ratio** quantifies the overhead or resource waste from padding. It is the ratio of the padded portion of a resource to the actual required resource size ($R_{\text{actual}}$), where $R_{\text{allocated}}$ is the total allocated resource. A lower ratio is better.

$$\text{Padding Ratio} = \frac{R_{\text{allocated}} - R_{\text{actual}}}{R_{\text{actual}}}$$

## C  BENCHMARK DETAILS

### C.1  DATASET STATISTICS

As visualized in Figure 4, our benchmark suite is composed of a diverse set of datasets, categorized into three primary scenarios: Long-Sequence (4.7%), Reasoning (2.3%), and Reinforcement Learning (RL) (93.0%). Detailed statistics for each dataset, including the count of unique prompts in the training, validation, and test splits, are provided in Table 4. All datasets are partitioned into training, validation, and test sets following a 3:1:1 ratio.

Specifically, the Long-Sequence scenario includes:

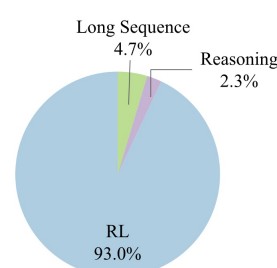

Figure 4: Proportional distribution of unique prompts across the three

Table 4: Statistics of the benchmark suite. Numbers indicate the count of **unique prompts** in each split.

| Dataset | Scenario | Train | Validation | Test | Total |
|---|---|---|---|---|---|
| LongBench | Long-Sequence | 330 | 110 | 110 | 550 |
| ZeroSCROLLS | Long-Sequence | 330 | 110 | 110 | 550 |
| IFEval | Reasoning | 330 | 110 | 110 | 550 |
| CRUXEval | RL | 480 | 160 | 160 | 800 |
| GSM8K | RL | 4,483 | 1,494 | 1,494 | 7,471 |
| LiveCodeBench | RL | 633 | 211 | 211 | 1,055 |
| MATH | RL | 4,500 | 1,500 | 1,500 | 7,500 |
| MBPP | RL | 1,157 | 386 | 386 | 1,929 |
| MMLU-STEM | RL | 1,891 | 630 | 630 | 3,151 |
| **Total** | | **14,134** | **4,711** | **4,711** | **23,556** |

- **LongBench** (Bai et al., 2024a) is a bilingual, multi-task benchmark designed to assess the understanding of long text contexts. It encompasses a variety of tasks such as single and multi-document question answering, summarization, and code completion.
- **ZeroSCROLLS** (Shaham et al., 2023) provides a suite of datasets focused on zero-shot evaluation of long-text comprehension. It includes tasks that require synthesizing information across lengthy documents, such as summarization, question answering, and sentiment classification.

The Reasoning scenario contains:

- **IFEval** (Zhou et al., 2023) is a dataset consisting of prompts with explicit and verifiable instructions. It is used to evaluate a model's ability to adhere to constraints and follow complex directives.

The RL scenario includes:

- **CRUXEval** (Gu et al., 2024) is a dataset for evaluating code reasoning, understanding, and execution. It tests a model's ability to predict the output of code snippets and to determine the necessary input to achieve a desired output.
- **LiveCodeBench** (Jain et al., 2025) is a dataset for code generation that features problems from competitive programming websites, focusing on problem-solving with varying levels of difficulty.
- **MBPP** (Austin et al., 2021) is a dataset containing entry-level Python programming problems that can be solved with short, self-contained functions.
- **GSM8K** (Cobbe et al., 2021) is a dataset of grade school math word problems that require multi-step reasoning to solve.
- **MATH** (Hendrycks et al., 2021b) is a dataset of challenging mathematical problems that require sophisticated reasoning and problem-solving abilities.
- **MMLU-STEM** (Hendrycks et al., 2021a) is a subset of the Massive Multitask Language Understanding benchmark that focuses on science, technology, engineering, and mathematics subjects, designed to test a model's knowledge and problem-solving skills in these areas.

## C.2 PROMPT AND RESPONSE LENGTH DISTRIBUTIONS

As detailed in Table 5 and visualized in Figure 5 and Figure 6, our benchmark features a wide diversity of sequence lengths. Prompt lengths range from the short, concise queries in reasoning

datasets like MATH and GSM8K, which average a few hundred characters, to the extensive contexts in long-sequence datasets like LongBench and ZeroSCROLLS, which average nearly 6,000 characters. Response lengths are similarly varied, reflecting the diverse complexity of the tasks that demand outputs ranging from single-word answers to detailed reasoning and comprehensive code solutions.

Table 5: Statistical summary of prompt and response lengths across all datasets. The table presents the minimum, maximum, and mean length for each category.

| Dataset | Prompt Length | | | Response Length | | |
|---|---|---|---|---|---|---|
| | Min | Max | Mean | Min | Max | Mean |
| CRUXEval | 162 | 387 | 257 | 1 | 6,841 | 1,136 |
| LiveCodeBench | 572 | 4,089 | 1,547 | 1 | 7,288 | 2,188 |
| MMLU-STEM | 43 | 1,574 | 280 | 1 | 7,530 | 1,058 |
| MBPP | 107 | 319 | 153 | 16 | 6,412 | 594 |
| MATH | 16 | 4,309 | 210 | 2 | 7,579 | 736 |
| GSM8K | 42 | 985 | 235 | 11 | 8,063 | 598 |
| IFEval | 53 | 1,858 | 211 | 2 | 3,730 | 1,586 |
| LongBench | 624 | 7,994 | 5,865 | 3 | 7,106 | 1,172 |
| ZeroSCROLLS | 4,004 | 7,983 | 5,914 | 1 | 3,626 | 1,420 |

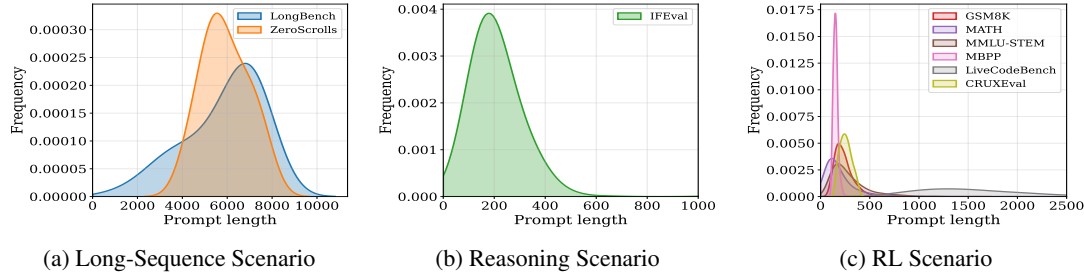

(a) Long-Sequence Scenario      (b) Reasoning Scenario      (c) RL Scenario

Figure 5: Prompt length distributions across different scenarios.

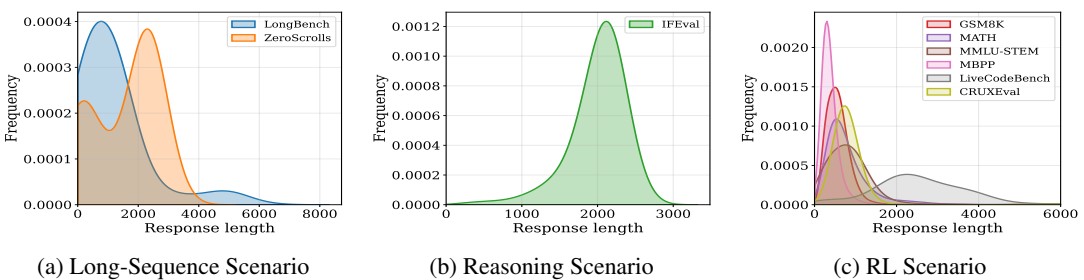

(a) Long-Sequence Scenario      (b) Reasoning Scenario      (c) RL Scenario

Figure 6: Response length distributions across different scenarios.

## C.3 BENCHMARK DATASET EXAMPLES

To provide a clearer understanding of our benchmark, this section presents representative examples from our three scenarios. The long-sequence example tests the model's ability to synthesize information from extensive contexts. The reasoning example requires multi-step logical deduction. Finally, the RL example is taken directly from our GRPO training data.

**An Example of Long-Sequence Scenario**

**[Prompt]**
Question: Who is Renaud II, Count Of Soissons's uncle?

Context:
Passage 1:
John I, Count of Soissons. John became Count of Soissons after the death of his brother Renaud II in 1099. John married Aveline de Pierrefonds, daughter of Nivelon II, Seigneur de Pierrefonds. They had: - Renaud III, Count of Soissons.
...

Passage 2:
Adelaide, Countess of Soissons. Adelaide was sovereign Countess of Soissons from 1057 until 1105. She was the daughter of Renaud I, Count of Soissons, and his wife, whose name is unknown. Adelaide married William Busac, Count of Eu. William became Count of Soissons in right of his wife.
...

Passage 5:
Renaud II, Count of Soissons. Renaud II became Count of Soissons after his father's disgrace. It is unclear when he assumed the countship, but after his death in 1099, his brother John became the Count of Soissons.
...

**[Response]**
Renaud II, Count of Soissons' uncle is William Busac, Count of Eu and Soissons.

---

**An Example of Reasoning Scenario**

**[Prompt]**
What is the next number in this series: 1, 4, 7, 11, 17? Please answer with only mathematical notation without any commas.
**[Response]**
```
<think>
```
To find the next number in the series 1, 4, 7, 11, 17, we first need to identify the pattern. We can do this by finding the differences between consecutive terms:
4 - 1 = 3
7 - 4 = 3
11 - 7 = 4
17 - 11 = 6
The differences between consecutive terms are increasing by 1 each time (3, 3, 4, 6). Following this pattern, the next difference should be: 6 + 2 = 8.
To find the next term in the sequence, we add this difference to the last term of the sequence: 17 + 8 = 25
```
</think>
<answer>
```
Thus, the next number in the series is 25.
```
</answer>
```

---

**An Example of RL Scenario**

**[Prompt]**
Write a function with the following signature: `def first_Factorial_Divisible_Number(x)`. Write a python function to find the first natural number whose factorial is divisible by x.

**[Response]**
```
<reasoning>
```
To find the first natural number whose factorial is divisible by x, I need to write a function which calculates the factorial of the numbers starting from 1 and checks if the resulting factorial is divisible by x. As soon as I find a number with this property, I return that number.
1. Initialize a variable n (starting from 1) to keep track of the current number.
2. Initialize a variable fact (starting from 1) to store the factorial value.
3. Loop through the numbers starting from 1 and calculate the factorial of each number.
4. Check if the factorial is divisible by x.
5. If it is, return the current number.
This approach should yield the first natural number whose factorial is divisible by x.
```
</reasoning>
<answer>

def first_Factorial_Divisible_Number(x):
    n = 1
    fact = 1
    while True:
        if fact % x == 0:
            return n
        n += 1
        fact *= n

</answer>
```

---

## C.4 System Prompts

In this section, we present the system prompts used to collect model responses. The design of each prompt is carefully tailored to the requirements of the corresponding dataset. For straightforward tasks like long-context question answering (LongBench, ZeroSCROLLS), we use a simple, direct prompt. For more complex tasks involving reasoning (IFEval) or coding (GSM8K, MATH, MMLU-STEM, MBPP, LiveCodeBench, CRUXEval), we employ more detailed and structured prompts. These structured formats, often requiring the model to expose its step-by-step reasoning within specific tags (e.g., `</reasoning>`), are crucial for improving the reliability of the model's output and enabling more accurate evaluation. The level of strictness in the prompt increases with the complexity and evaluation requirements of the task.

---

**System Prompt for IFEval Dataset**

You MUST respond in exactly this format:

```
<think>
```
[Write your step-by-step reasoning process here. Explain how you will approach the task, what you need to consider, and work through the problem systematically.]
```
</think>
<answer>
```
[Provide your final answer here based on your reasoning above.]

```
</answer>
```
Always use these exact tags `<think>` and `<answer>`. Do not skip them or use different formatting.

---

---

**System Prompt for GSM8K Dataset**

Respond in the following format:

```
<reasoning>
...
</reasoning>
<answer>
...
</answer>
```

---

**System Prompt for LongBench and ZeroSCROLLS Datasets**

You are an intelligent assistant capable of understanding and analyzing long contexts. Your task is to carefully read the provided context and answer the given question accurately and comprehensively.
Instructions:
1. Read the entire context carefully
2. Understand the question being asked
3. Provide a clear, accurate, and well-reasoned answer based on the context
4. If the question cannot be answered from the context, clearly state so
5. For multilingual content, respond in the same language as the question

---

**System Prompt for MATH Dataset**

You are a mathematical problem solver. Solve the given problem step by step with clear mathematical reasoning.
Guidelines:
1. Read the problem carefully and identify what is being asked
2. Identify the given information and any constraints
3. Choose the appropriate mathematical concepts, formulas, or methods
4. Show your work step by step with clear explanations 5. Perform calculations accurately
6. Verify your answer makes sense in the context of the problem
7. Present your final answer clearly

Respond in the following format:
```
<reasoning>
```
Step 1: [Identify what the problem is asking and what information is given]
Step 2: [Choose the mathematical approach/method to solve the problem]
Step 3: [Set up equations, formulas, or mathematical expressions]
Step 4: [Perform calculations step by step, showing all work]
Step 5: [Verify the solution and check if it makes sense]
```
</reasoning>
<answer>
```
[Provide the final numerical answer or mathematical expression. For numerical answers, give exact values when possible (fractions, radicals) or decimal approximations when appropriate. Clearly state units if applicable.]
```
</answer>
```

**System Prompt for MMLU-STEM Dataset**

Respond in the following format:
```
<reasoning>
```
[Provide your step-by-step reasoning here]
```
</reasoning>
<answer>
```
[Put only the number (1, 2, 3, or 4) of your chosen answer here]
```
</answer>
```
Do not add any extra text before or after the XML tags.

**System Prompt for MBPP Dataset**

You are a helpful coding assistant. You must respond in the exact format shown below.
IMPORTANT: Your response must strictly follow this XML format:
```
<reasoning>
```
[Your step-by-step reasoning here]
```
</reasoning>
<answer>
```
[Your Python code solution here]
```
</answer>
```
Do not include any text outside of these XML tags. Do not use markdown code blocks. Place the code directly inside the `<answer>` tags.

**System Prompt for LiveCodeBench Dataset**

You are a helpful coding assistant. You MUST respond in the exact XML format shown below.
CRITICAL: Your response must STRICTLY follow this XML format. Any deviation will result in failure:
MANDATORY REQUIREMENTS:
1. Use ONLY the XML tags `<reasoning>` and `<answer>`
2. Do NOT use markdown code blocks
3. Do NOT include any text outside the XML tags
4. Place your Python code directly inside `<answer>` tags without any formatting
5. Start your response immediately with `<reasoning>`
6. End your response immediately with `</answer>`
WRONG FORMATS (DO NOT USE):
- Any text before `<reasoning>`
- Any text after `</answer>`
- Missing XML tags

CORRECT FORMAT EXAMPLE:
```
<reasoning>
```
I need to solve this step by step...
```
</reasoning>
</answer>

def solution():
    return "result"

</answer>
```

**System Prompt for CRUXEval Dataset**

You are an expert Python code execution simulator. Your task is to carefully trace through Python function execution step-by-step and predict the exact output.
When given a Python function and its input:
1. Carefully read and understand the function logic
2. Trace through each line of execution with the given input
3. Track variable states and transformations
4. Predict the final return value with precise formatting
Important guidelines:
- Pay attention to data types (lists, tuples, strings, numbers, booleans)
- Consider edge cases and special Python behaviors
- Maintain exact formatting for complex data structures
- For strings, preserve quotes and escape characters
- For None values, output exactly "None"
- For boolean values, output exactly "True" or "False"
Respond in the following format:
```
<reasoning>
```
Step-by-step execution trace explaining how you arrived at the answer
```
</reasoning>
<answer>
```
The exact output value that the function will return
```
</answer>
```

## D ABLATION STUDY

### D.1 LOSS WEIGHTING PARAMETER

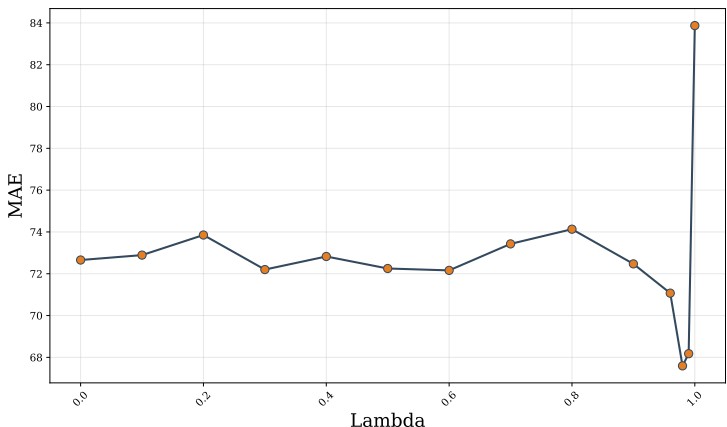

Figure 7: Ablation study on the loss weighting parameter $\lambda$.

We conduct a ablation study on LMSYS dataset to investigate the impact of the weighting parameter $\lambda$ in our joint loss function 9. As shown in Figure 7, the choice of $\lambda$ significantly affects model performance on the length prediction task. When using pure MSE loss ($\lambda = 0$), the model achieves the worst performance with MAE of approximately 82-84, which can be attributed to the inherent numerical scale disparity between MSE and cross-entropy losses—MSE values are typically two orders of magnitude larger than cross-entropy values in this high-complexity prediction task, leading to gradient domination issues. Pure cross-entropy loss ($\lambda = 1$) provides a solid baseline performance with MAE around 70, demonstrating the effectiveness of the classification-based approach. However, the optimal performance is achieved at $\lambda = 0.99$, yielding MAE of approximately 68, which represents a notable improvement over both extreme cases. This optimal weighting allows the cross-entropy loss to provide stable learning signals and handle the discrete nature of the prediction task, while the small contribution from MSE loss (1%) offers fine-grained regression-based optimization

for enhanced precision. The results validate our design choice and demonstrate that careful balance between complementary loss functions is crucial for achieving superior performance in complex prediction tasks.

## D.2 ANALYSIS OF LOSS FUNCTION VARIANTS

While our proposed EGTP employs a soft label distribution combined with regression, several alternative strategies exist for handling regression targets in length prediction. To validate the necessity and effectiveness of our design choice, we compared EGTP against three common alternatives:

- **Log-scale Regression:** Predicting the logarithm of the length ($\log(y)$) to compress the target space and mitigate the impact of long-tailed distributions.
- **Label Smoothing:** Applying uniform smoothing to the one-hot classification targets to prevent overconfidence and improve generalization.
- **Label Noise:** Adding small random noise to the regression targets during training to improve robustness.

We conducted experiments using the Qwen2.5-7B model across all three scenarios: Long-Sequence (LongSeq), Reasoning, and Reinforcement Learning (RL). The results, measured in Mean Absolute Error (MAE), are presented in Table 6.

Table 6: Performance comparison (MAE ↓) of EGTP against alternative loss function designs on Qwen2.5-7B. Best results are bolded.

| Method | LongSeq | Reasoning | RL |
|---|---|---|---|
| **EGTP (Ours)** | $\mathbf{83.65 \pm 1.28}$ | $139.14 \pm 5.13$ | $\mathbf{92.78 \pm 2.32}$ |
| Label Smoothing | $158.44 \pm 0.73$ | $144.93 \pm 19.60$ | $268.88 \pm 3.20$ |
| Log-scale Regression | $135.73 \pm 2.04$ | $\mathbf{136.09 \pm 4.76}$ | $208.02 \pm 0.50$ |
| Label Noise | $119.05 \pm 2.25$ | $217.44 \pm 1.81$ | $201.77 \pm 1.91$ |

The results show that while Log-scale regression is competitive on the Reasoning task, EGTP consistently outperforms others across all three domains, especially in LongSeq and RL, indicating better robustness.

## D.3 LAYER SELECTION STRATEGY

The practice of using final-layer representations for transfer learning, common in architectures like VGG and ResNet, persists in modern LLMs. Notably, systems such as Retrieval-Augmented Generation (RAG) (Li et al., 2025b) employ these activations for retrieval and classification, validating their semantic richness. Following this established paradigm, we rely on final-layer activations to perform length prediction.

To validate the rationale of using the final layer, we conducted experiments on the GSM8k dataset using hidden states from different layers of Qwen2.5-0.5B (which has 24 layers) to predict output length. As shown in Table 3, the performance of the final layer (Layer 24) is comparable to that of the best-performing intermediate layers (such as Layer 12 and Layer 16). Layer 24 achieves an RMSE of 92.95 ± 2.80, which is close to Layer 12's 93.47 ± 1.50. Although Layer 12 slightly outperforms in terms of MAE (70.26 ± 1.20), Layer 24's MAE (73.93 ± 2.10) still maintains a competitive level. Considering that using the final layer avoids over-engineering and unnecessary implementation complexity, we consistently utilize the hidden states from the last transformer layer across all model types and sizes in our final implementation.

## E SUPPLEMENTAL EXPERIMENT RESULTS

### E.1 BASELINES

We use the following methods as our baselines in the main experiment.

Table 7: Performance of using different layers on Qwen2.5-0.5B (GSM8K). Metrics reported are RMSE and MAE (lower is better ↓).

| Layer Index | RMSE ↓ | MAE ↓ |
|---|---|---|
| Layer 1 | $110.18 \pm 2.50$ | $85.91 \pm 1.82$ |
| Layer 8 | $97.45 \pm 1.81$ | $81.29 \pm 1.50$ |
| Layer 12 | $93.47 \pm 1.50$ | $\mathbf{70.26 \pm 1.20}$ |
| Layer 16 | $94.30 \pm 1.64$ | $70.79 \pm 1.36$ |
| Layer 24 (Final) | $\mathbf{92.95 \pm 2.80}$ | $73.93 \pm 2.10$ |

- **SSJF** (Qiu et al., 2024): Uses a fine-tuned BERT model, formulated either as a regression task to predict the absolute token length (**SSJF-Reg**) or as a multi-class classification task for length categories (**SSJF-MC**).
- **S3** (Jin et al., 2023): Utilizes a fine-tuned Distilbert model to classify the output into one of ten predefined length buckets.
- **PiA** (Zheng et al., 2023): Prompts or instruction-tunes a Vicuna model to predict its own response length.
- **TPV** (Eisenstadt et al., 2025): Trains a linear regressor to estimate the relative progress of DeepSeek-R1-Distill's reasoning process.
- **TRAIL** (Shahout et al., 2025): Trains an MLP classifier on the Llama's internal embeddings to predict the final output length.
- **LTR** (Fu et al., 2024): Employs an OPT model backbone, trained through classification (**LTR-C**) or ranking, to predict the output sequence length.

## E.2 GENERALIZATION TO SUPER-LONG SEQUENCES

Most standard Supervised Fine-Tuning (SFT) datasets predominantly contain sequences shorter than 4k tokens (Bai et al., 2024b). Consequently, our main experiments focus on these typical distributions. However, to investigate the generalization capability of EGTP on Out-of-Distribution (OOD) samples with extreme lengths, we conduct additional experiments on the `euclaise/writingprompts` (Fan et al., 2018) dataset, where the longest sequence exceeds 17k tokens.

Table 8: Performance comparison on super-long sequences from the `euclaise/writingprompts` dataset (> 17k tokens). EGTP demonstrates superior generalization capability in this OOD setting.

| Metric | EGTP | SSJF-Reg | SSJF-MC | S3 | PiA | TPV | TRAIL | LTR-C |
|---|---|---|---|---|---|---|---|---|
| MAE ↓ | $\mathbf{195.89} \pm 1.54$ | $280.26 \pm 6.63$ | $315.25 \pm 16.88$ | $211.02 \pm 27.95$ | $214.48 \pm 5.15$ | $724.04 \pm 16.80$ | $212.73 \pm 6.00$ | $255.12 \pm 20.46$ |
| RMSE ↓ | $\mathbf{257.49} \pm 43.18$ | $366.32 \pm 69.61$ | $418.76 \pm 19.52$ | $281.78 \pm 40.07$ | $283.10 \pm 13.74$ | $1049.21 \pm 143.57$ | $279.42 \pm 4.22$ | $293.27 \pm 11.18$ |

The results in Table 8 demonstrate that EGTP maintains strong generalization even on sequences far exceeding the length distribution seen during training. It achieves the lowest MAE (195.89) and RMSE (257.49), significantly outperforming baselines such as TPV and SSJF-MC which fail to scale effectively. This suggests that the entropy-guided representations capture intrinsic autoregressive patterns related to generation termination, rather than merely memorizing length biases from the training set.

## E.3 COMPUTATIONAL EFFICIENCY ANALYSIS

To offer a more granular quantitative analysis of the overhead introduced by different length prediction methods, we evaluated the inference latency and memory consumption of the prediction modules in isolation. This complements the end-to-end system performance reported in Section 4.4.

We utilized the GSM8K math reasoning dataset for evaluation. All experiments were conducted on a single NVIDIA RTX 4090 (24GB) GPU under consistent environmental settings. We measured:

- **Avg Time (ms):** The average wall-clock time required for the predictor to process a query and generate a length estimate.
- **Avg VRAM (MB):** The additional GPU memory required to load and run the prediction module.

We compared EGTP against baselines. Note that for PiA, which relies on the LLM itself to generate token counts, we exclusively recorded the time consumed for generating the specific length-prediction tokens to ensure a fair comparison.

Table 9: Auxiliary model overhead for predicting **Qwen2.5-7B** response lengths. EGTP incurs negligible latency and memory costs compared to methods requiring external models.

| Metric | EGTP (Ours) | SSJF-MC | SSJF-Reg | S3 | LTR-C | TPV | PiA | TRAIL |
|---|---|---|---|---|---|---|---|---|
| Avg Time (ms) | $0.67 \pm 0.07$ | $3.94 \pm 0.01$ | $4.04 \pm 0.12$ | $2.26 \pm 0.08$ | $12.57 \pm 0.07$ | $0.83 \pm 0.21$ | $60.90 \pm 0.33$ | $2.04 \pm 0.08$ |
| Avg VRAM (MB) | 7.21 | 269.76 | 269.76 | 264.19 | 238.41 | 6.38 | - | 268.03 |

Table 10: Auxiliary model overhead for predicting **Llama3.2-3B** response lengths.

| Metric | EGTP (Ours) | SSJF-MC | SSJF-Reg | S3 | LTR-C | TPV | PiA | TRAIL |
|---|---|---|---|---|---|---|---|---|
| Avg Time (ms) | $0.65 \pm 0.01$ | $3.95 \pm 0.02$ | $4.18 \pm 0.13$ | $2.41 \pm 0.04$ | $12.56 \pm 0.03$ | $0.79 \pm 0.11$ | $60.40 \pm 1.97$ | $2.59 \pm 0.11$ |
| Avg VRAM (MB) | 5.52 | 269.76 | 269.76 | 264.19 | 238.41 | 5.95 | - | 268.03 |

Tables 9 and 10 unequivocally demonstrate the efficiency of our approach. EGTP achieves the lowest inference time ($\approx 0.66$ms), which is effectively negligible in the context of LLM inference. In contrast, auxiliary model-based methods like SSJF and TRAIL typically require 2–12ms to perform a forward pass on their external models. PiA is the slowest ($\approx 60$ms) due to the overhead of autoregressive decoding for the prediction tokens.

Since EGTP reuses the hidden states already computed during the prefill phase of the main LLM, it only requires storing a lightweight linear head, consuming merely 5–7 MB of VRAM. Conversely, baselines requiring separate auxiliary models consume significantly more memory (238–270 MB) to store model weights, which can compete for resources with the main serving system.

### E.4 LENGTH PREDICTION PERFORMANCE ON OTHER DATASETS

Table 11 presents a comprehensive comparison of our proposed method, EGTP, against several baseline approaches using the Qwen2.5 model family (0.5B and 1.5B). The evaluation, based on Mean Absolute Error (MAE), is conducted across three distinct scenarios: Long Sequence, Reasoning, and RL. The results clearly demonstrate the superiority of our method. EGTP consistently achieves the lowest average MAE for both model sizes, significantly outperforming all competitors. The TRAIL method consistently secures the second-best position, but still trails our approach by a notable margin.

| Benchmark | | Prediction Method | | | | | | | |
|---|---|---|---|---|---|---|---|---|---|
| Model | Scenario | EGTP (Ours) | SSJF-Reg | SSJF-MC | S3 | PiA | TPV | TRAIL | LTR-C |
| **Qwen2.5 0.5B** | Long Seq | 133.10 | 375.67 | 675.28 | 380.45 | 444.17 | 847.42 | 170.74 | 215.44 |
| | Reasoning | 145.85 | 221.15 | 466.00 | 200.91 | 296.02 | 597.00 | 146.24 | 178.28 |
| | RL | 88.52 | 125.40 | 224.68 | 179.19 | 255.52 | 261.36 | 161.79 | 199.14 |
| | **Avg.** | **122.49** | 240.74 | 455.32 | 253.52 | 331.90 | 568.59 | **159.59** | 197.62 |
| **Qwen2.5 1.5B** | Long Seq | 125.42 | 301.68 | 690.73 | 212.73 | 351.20 | 587.64 | 151.86 | 162.54 |
| | Reasoning | 146.48 | 310.64 | 815.78 | 169.72 | 432.05 | 584.08 | 138.01 | 139.02 |
| | RL | 96.59 | 129.65 | 173.95 | 167.97 | 133.82 | 208.39 | 160.60 | 170.93 |
| | **Avg.** | **122.83** | 247.32 | 560.15 | 183.47 | 305.69 | 460.04 | **150.16** | 157.50 |

Table 11: Comparison of different length prediction methods based on Mean Absolute Error (MAE). Lower values indicate better performance. In each 'Avg.' row, which shows the mean performance, the **best-performing** method is highlighted in red, and the **second-best** is highlighted in green.

E.5 DETAILED PREDICTION ACCURACY RESULTS

The following tables provide a detailed breakdown of the output length prediction results, supplementing the averaged scores presented in the main text. We evaluate several baseline methods on their ability to predict the response length from a given prompt. The prompts are sourced from Long-Bench, ZeroSCROLLS (See Table 12 and Table 13), and IFEval (See Table 14 and Table 15). The corresponding responses are generated by the LLMs specified in the "Model" column (Qwen2.5, Llama3.2, and DeepSeek-R1-Distill models).

Table 12: Performance of length prediction methods on Long-Sequence Scenario.

| Model | Method | LongBench | ZeroSCROLLS |
|---|---|---|---|
| | | MAE | MAE |
| Qwen2.5 -0.5B -Instruct | SSJF-Reg | 374.14 | 140.36 |
| | SSJF-MC | 696.02 | 654.55 |
| | S3 | 599.09 | 161.82 |
| | PiA | 325.00 | 146.50 |
| | TPV | 1078.33 | 616.52 |
| | TRAIL | **176.19** | 165.29 |
| | LTR-C | 339.78 | **91.09** |
| | **EGTP(Ours)** | **114.42** | **95.44** |
| Qwen2.5 -1.5B -Instruct | SSJF-Reg | 250.71 | **92.07** |
| | SSJF-MC | 720.45 | 660.99 |
| | S3 | 300.91 | 124.55 |
| | PiA | 214.32 | 151.48 |
| | TPV | 705.29 | 469.98 |
| | TRAIL | **147.95** | 155.77 |
| | LTR-C | 207.55 | 117.53 |
| | **EGTP(Ours)** | **93.36** | **82.74** |

Table 13: Performance of length prediction methods on Long-Sequence Scenario (MAE only).

| Model | Method | LongBench MAE | ZeroSCROLLS MAE |
|---|---|---|---|
| Qwen2.5 -3B -Instruct | SSJF-Reg | 208.55 | **96.75** |
| | SSJF-MC | 930.60 | 613.32 |
| | S3 | 254.55 | 119.09 |
| | PiA | 153.00 | 161.45 |
| | TPV | 790.71 | 361.45 |
| | TRAIL | **143.47** | 152.36 |
| | LTR-C | 153.51 | 94.95 |
| | **Ours** | **132.43** | **78.83** |
| Qwen2.5 -7B -Instruct | SSJF-Reg | 201.49 | 100.93 |
| | SSJF-MC | 832.60 | 182.08 |
| | S3 | 223.64 | **100.00** |
| | PiA | 256.32 | 154.24 |
| | TPV | 676.28 | 391.57 |
| | TRAIL | **150.46** | 117.91 |
| | LTR-C | 158.45 | 100.29 |
| | **Ours** | **114.97** | **93.43** |

**Table 13 – continued from previous page**

| Model | Method | LongBench MAE | ZeroSCROLLS MAE |
|-------|--------|---------------|------------------|
| Llama3.2 -1B -Instruct | SSJF-Reg | 290.54 | **57.40** |
| | SSJF-MC | 431.19 | 72.24 |
| | S3 | 450.91 | 73.64 |
| | PiA | **145.61** | 142.41 |
| | TPV | 686.38 | 337.95 |
| | TRAIL | 161.89 | 128.82 |
| | LTR-C | 307.63 | **50.59** |
| | **Ours** | **93.36** | 59.19 |
| Llama3.2 -3B -Instruct | SSJF-Reg | 311.58 | **75.09** |
| | SSJF-MC | 387.15 | 82.50 |
| | S3 | 441.82 | **77.27** |
| | PiA | **142.32** | 145.61 |
| | TPV | 1079.16 | 388.35 |
| | TRAIL | 172.45 | 114.79 |
| | LTR-C | 504.68 | 131.10 |
| | **Ours** | **102.47** | 79.03 |

Table 14: Performance of length prediction methods on Reasoning Scenario.

| Model | Method | IFeval MAE |
|-------|--------|------------|
| Qwen2.5 -0.5B -Instruct | SSJF-Reg | **149.27** |
| | SSJF-MC | 483.46 |
| | S3 | 200.92 |
| | PiA | 260.06 |
| | TPV | 597.00 |
| | TRAIL | 146.24 |
| | LTR-C | 178.28 |
| | **Ours** | **145.81** |
| Qwen2.5 -1.5B -Instruct | SSJF-Reg | **133.99** |
| | SSJF-MC | 597.21 |
| | S3 | 180.73 |
| | PiA | 265.15 |
| | TPV | 584.08 |
| | TRAIL | **138.01** |
| | LTR-C | 139.02 |
| | **Ours** | 146.48 |
| Qwen2.5 -3B -Instruct | SSJF-Reg | **129.00** |
| | SSJF-MC | 812.80 |
| | S3 | 223.85 |
| | PiA | 254.57 |
| | TPV | 621.72 |
| | TRAIL | 132.19 |
| | LTR-C | 145.53 |
| | **Ours** | **111.23** |
| Qwen2.5 -7B -Instruct | SSJF-Reg | **120.52** |
| | SSJF-MC | 599.25 |

**Table 14 – continued from previous page**

| Model | Method | MAE |
|-------|--------|-----|
|       | S3     | 168.81 |
|       | PiA    | 254.54 |
|       | TPV    | 466.56 |
|       | TRAIL  | 124.19 |
|       | LTR-C  | 134.55 |
|       | **Ours** | **119.60** |

Table 15: Performance of length prediction methods on Reasoning Scenario.

| Model | Method | MAE |
|---|---|---|
| DeepSeek-R1-Distill -Qwen-1.5B | SSJF-Reg | 86.16 |
| | SSJF-MC | 90.36 |
| | S3 | 164.22 |
| | PiA | 264.66 |
| | TPV | 373.03 |
| | TRAIL | 77.16 |
| | LTR-C | **74.01** |
| | **Ours** | **71.59** |
| DeepSeek-R1-Distill -Qwen-7B | SSJF-Reg | 57.67 |
| | SSJF-MC | 143.78 |
| | S3 | 52.29 |
| | PiA | 254.57 |
| | TPV | 273.04 |
| | TRAIL | 58.28 |
| | LTR-C | **52.28** |
| | **Ours** | **48.36** |
| DeepSeek-R1-Distill -Llama-8B | SSJF-Reg | 72.00 |
| | SSJF-MC | 78.21 |
| | S3 | 69.45 |
| | PiA | 272.16 |
| | TPV | 243.54 |
| | TRAIL | **46.85** |
| | LTR-C | 66.06 |
| | **Ours** | **45.87** |

## E.6 VISUALIZATION OF THE GRPO TRAINING PROCESS

To demonstrate the effectiveness of our GRPO training process, we present the reward curves from our experiments in Figure 8 through Figure 13. These visualizations cover the training process on six datasets: code execution prediction with CRUXEval (Figure 8), code generation with MBPP (Figure 9) and LiveCodeBench (Figure 12), and mathematical and scientific reasoning with MMLU-STEM (Figure 10), MATH (Figure 11), and GSM8K (Figure 13).

For each dataset, we trained multiple models, including four different sizes of Qwen2.5 and two sizes of Llama3.2. It should be noted that for smaller models, we use a larger batch size, which can result in a different number of training steps. As shown across all figures, the training process exhibits a stable and consistent improvement in rewards. This demonstrates that our GRPO implementation successfully optimizes model performance, regardless of the underlying architecture or the specific challenges of the task.

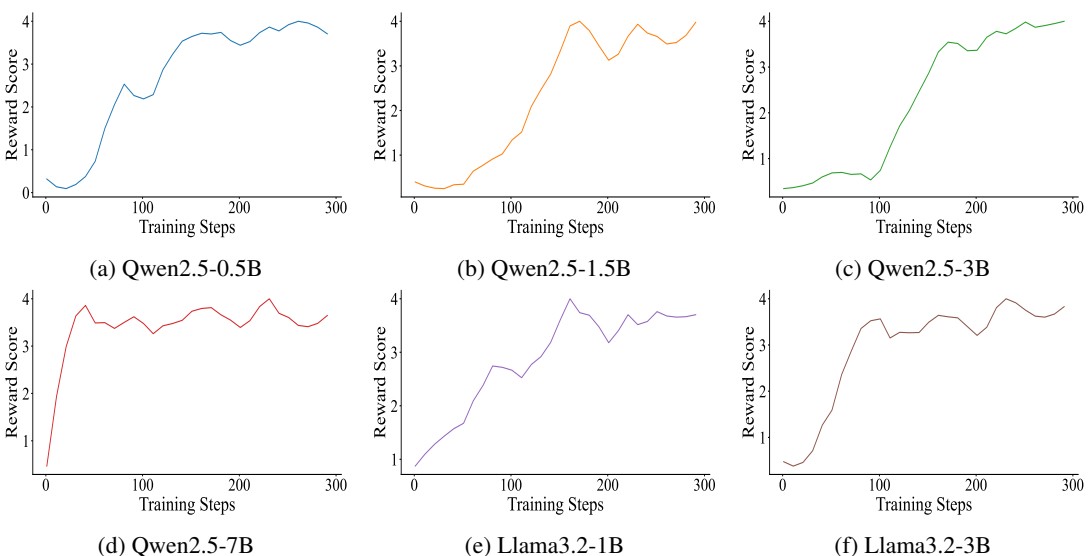

Figure 8: GRPO training process on the CRUXEval dataset.

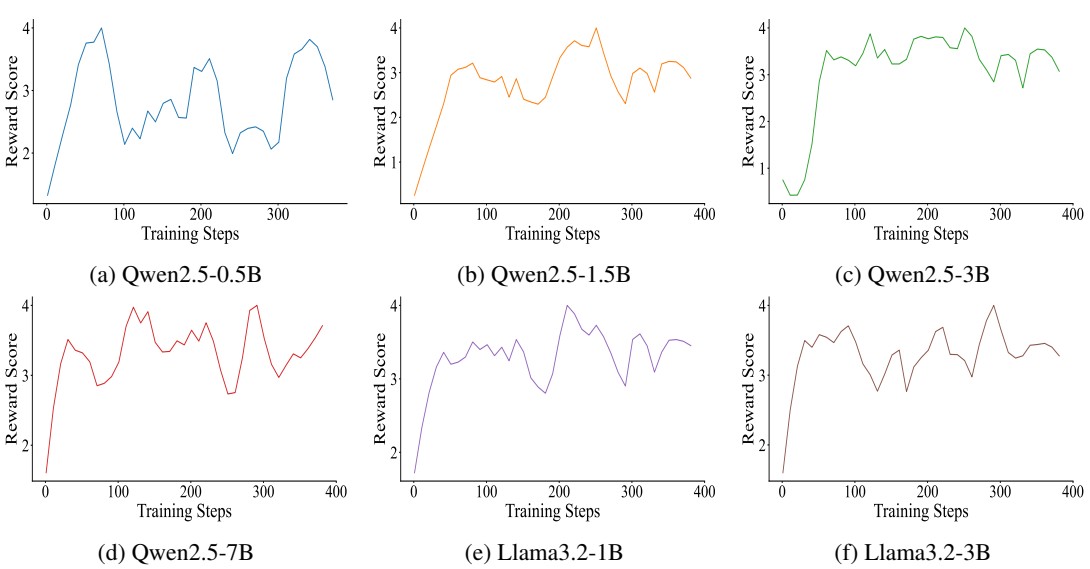

Figure 9: GRPO training process on the MBPP dataset.

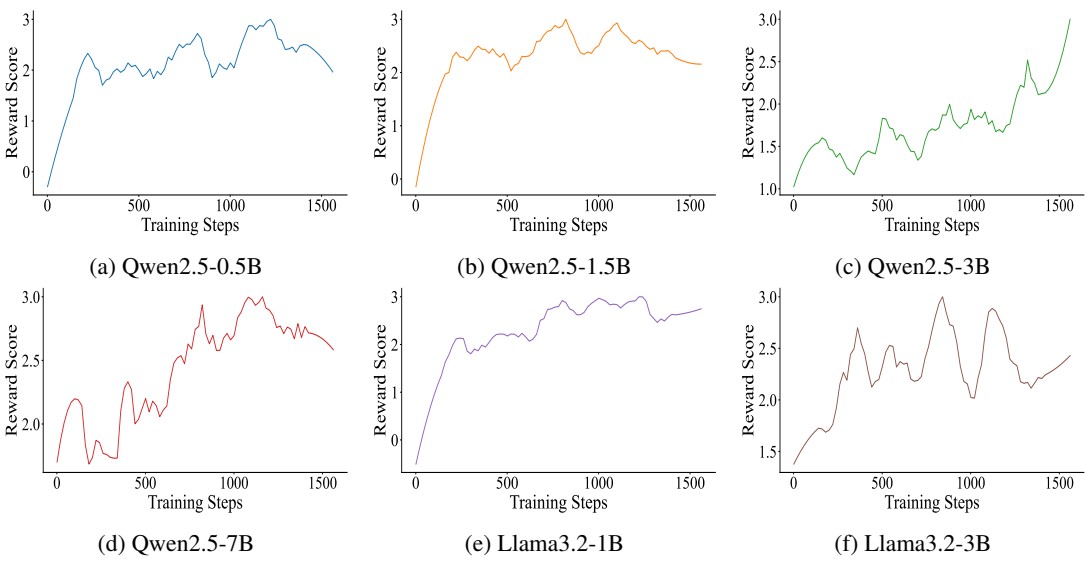

Figure 10: GRPO training process on the MMLU-STEM dataset.

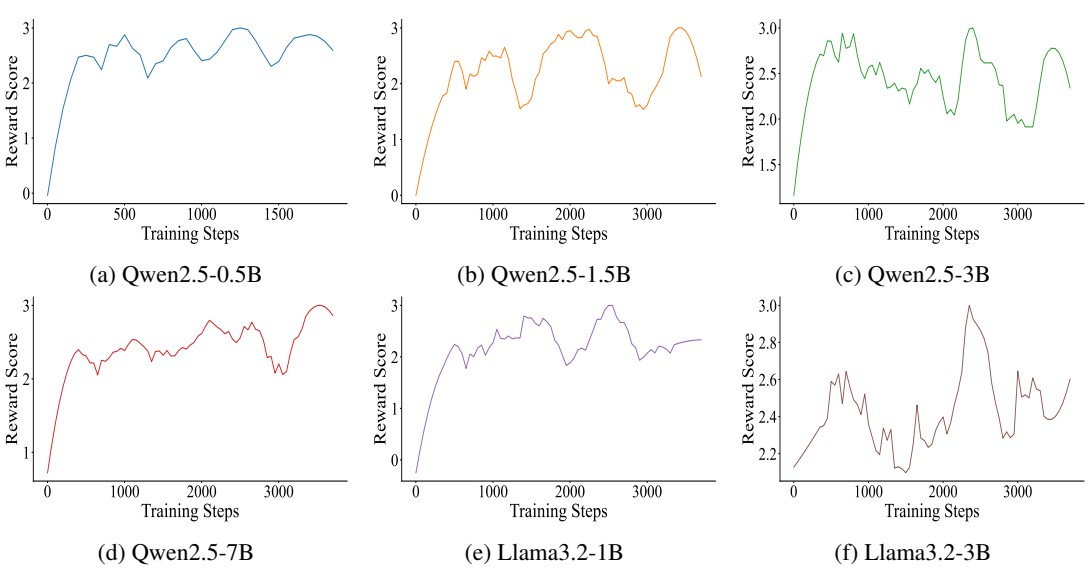

Figure 11: GRPO training process on the MMLU-STEM dataset.

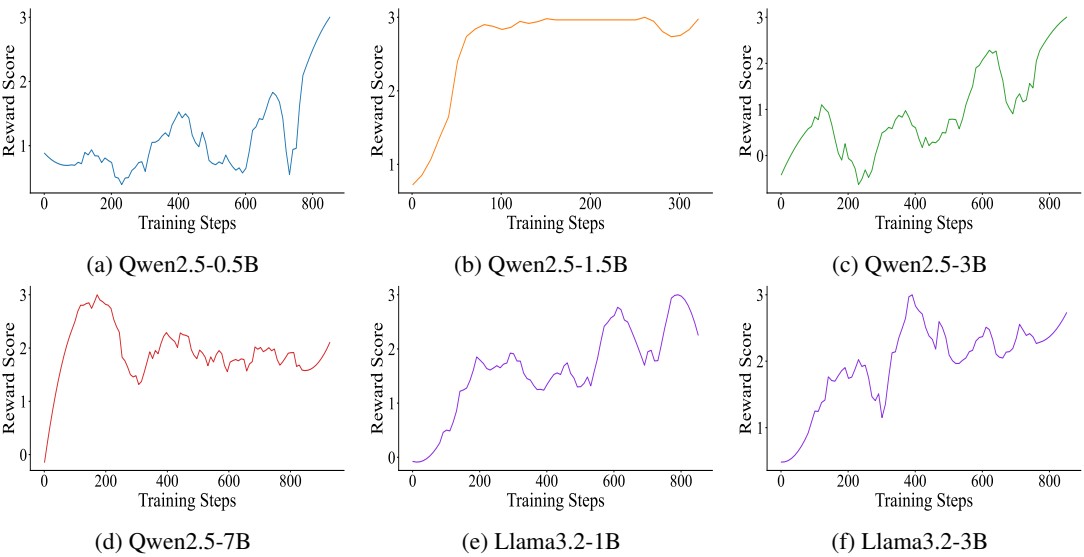

Figure 12: GRPO training process on the LiveCodeBench dataset.

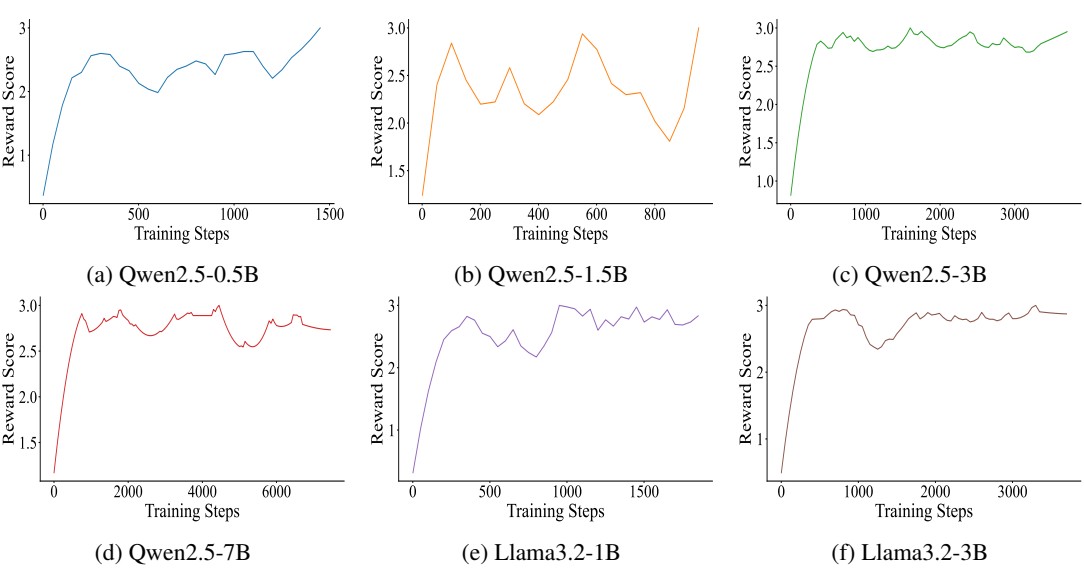

Figure 13: GRPO training process on the GSM8K dataset.

