# OpenReview forum: "Predicting LLM Output Length via Entropy-Guided Representations"
_ICLR.cc/2026/Conference — ICLR 2026 Poster_

### Official Review · Reviewer_Reek · 2025-10-20

**Soundness:** 3
**Presentation:** 3
**Contribution:** 3
**Rating:** 6
**Confidence:** 3

**Summary:**

This paper addresses the problem of predicting LLM output lengths by two core components: 1) Entropy-Guided Token Pooling (EGTP), which uses on-the-fly activations and token entropy for highly accurate static prediction with negligible cost, and 2) Progressive Length Prediction (PLP), which dynamically estimates the remaining length at each decoding step to handle stochastic generation.

The paper is pretty interesting with some well-designed experiments. The new benchmark ForeLen is sound, which covers long-sequence, Chain-of-Thought, and RL sampling data. I believe this benchmark would help the community, and I am looking forward to seeing it open-source.

**Strengths:**

The paper clearly articulates why existing auxiliary predictor methods are problematic (overhead, poor generalization, stochastic generation failures) and proposes a principled alternative.

The performance of predicting LLM output is important for real-world application. We are glad to see the work outperform other predicting methods.

ForeLen addresses limitations of LMSYS by including longer sequences and RL data, which will benefit future research

**Weaknesses:**

This paper uses the Mean Absolute Error (MAE) to evaluate the predictor's performance. Further analysis of statistical parameters can help readers understand the advantages and disadvantages of different methods, including variance and mean squared error.

The core insight (reusing internal representations) is somewhat incremental over prior work like TRAIL that also uses embeddings.

I think we can even predict the full output if we extract the model's internal activations. That makes this work make sense and also limits the scope of the potential benefit. I am looking forward to seeing the following work.

**Questions:**

How does EGTP perform on very long sequences beyond the training distribution (e.g., >10k tokens)?

---

> ### Author Response · Authors · 2025-11-23
>
> > This paper uses the Mean Absolute Error (MAE) to evaluate the predictor's performance. Further analysis of statistical parameters can help readers understand the advantages and disadvantages of different methods, including variance and mean squared error.
>
> MAE is a widely used metric for length prediction and regression tasks, which we adopt to ensure fair comparison with baseline methods. Following the reviewer's suggestion, we also supplement our evaluation with additional statistical metrics (RMSE and prediction variance) for Qwen2.5-7B, providing a more comprehensive assessment of each method's performance characteristics.
>
> Table 1: RMSE comparison across different task categories on Qwen2.5-7B (lower is better ↓).
> | Method | Long Seq | Reasoning | RL |
> | :--- | :---: | :---: | :---: |
> | **EGTP (Ours)** | 108.47+-2.43 | 123.25+-4.32 | 120.46+-1.33 |
> | SSJF-Reg | 313.64±7.84 | 332.10±15.95 | 306.85±12.05 |
> | SSJF-MC | 746.19±25.33 | 1102.99±17.91 | 368.69±5.99 |
> | S3  | 174.25±2.15 | 203.66±9.09 | 192.69±6.56 |
> | PiA | 180.75±6.93 | 186.96±2.02 | 172.75±8.43 |
> | TPV | 597.57±25.87 | 679.05±12.56 | 240.92±4.13 |
> | TRAIL | 256.46±4.45 | 192.05±4.26 | 134.09±4.16 |
> | LTR-C | 168.26±4.59 | 206.47±4.47 | 140.86±4.86 |
>
> The results demonstrate that EGTP consistently achieves the lowest RMSE across all three scenarios with Qwen2.5-7B, showing particularly strong performance on long sequences (108.47) and maintaining competitive results on reasoning (123.25) and RL tasks (120.46). The low variance across runs indicates stable performance. Methods like LTR-C and TRAIL show reasonable performance on certain tasks but exhibit higher errors on long sequences, while SSJF-MC struggles significantly on reasoning tasks. This validates that our entropy-guided approach generalizes well across different model architectures and task types.
>
> > The core insight (reusing internal representations) is somewhat incremental over prior work like TRAIL that also uses embeddings. I think we can even predict the full output if we extract the model's internal activations. That makes this work make sense and also limits the scope of the potential benefit. I am looking forward to seeing the following work.
>
>
> We would like to clarify a key distinction: TRAIL employs a two-stage process that first uses DistilBERT for initial predictions and then refines them using LLM activations. In contrast, our method leverages only the LLM's internal activations guided by entropy, eliminating the need for an auxiliary language model. Beyond this architectural difference, our main contributions focus on entropy-guided selection and handling stochastic scenarios, which shows better performance than TRAIL in most scenarios.
>
> > How does EGTP perform on very long sequences beyond the training distribution (e.g., >10k tokens)?
>
> Due to computational costs and practical constraints, outputs exceeding 10k tokens are relatively rare in real-world applications. As reported in LongWriter [1], common SFT datasets predominantly contain sequences shorter than 4k tokens, which is why our main experiments focus on these typical dataset distributions.
>
> However, to investigate EGTP's generalization capability on super-long sequences, we conducted experiments on `euclaise/writingprompts` [2] to compare performance on super long sequences, where the longest sequence exceeds 17k tokens. Models were trained on the training set and evaluated on the test set. The results are shown as follows:
>
> Table 2: Performance comparison on super-long sequences. (lower is better ↓)
> |  | EGTP (Ours) | SSJF-Reg | SSJF-MC | S3 | PiA | TPV | TRAIL | LTR-C |
> | --- | --- | --- | --- | --- | --- | --- | --- | --- |
> | MAE | 195.89 ± 1.54 | 280.26 ± 6.63  | 315.25 ± 16.88 | 211.02 ± 27.95 | 214.48 ± 5.15 | 724.04 ± 16.80 | 212.73 ± 6.00 | 255.12 ± 20.46 |
> | RMSE | 257.49 ± 43.18 | 366.32 ± 69.61 | 418.76 ± 19.52 | 281.78 ± 40.07 | 283.10 ± 13.74 | 1049.21 ± 143.57 | 279.42 ± 4.22 | 293.27 ± 11.18 |
>
> The results demonstrate that EGTP maintains strong generalization on very long sequences, achieving the lowest MAE (195.89) and RMSE (257.49) compared to baselines. The entropy-guided approach appears to effectively capture length patterns even in out-of-distribution scenarios,  demonstrating strong generalization capability on super-long sequences.
>
> [1] LONGWRITER: UNLEASHING 10,000+ WORD GENERATION FROM LONG CONTEXT LLMS
>
> [2] Hierarchical Neural Story Generation

---

> ### Author Response · Authors · 2025-11-23
>
> We thank the reviewer for the constructive feedback. We have carefully addressed the raised concerns and hope this clarification may support a more favorable reassessment.

---

> ### Author Response · Authors · 2025-11-24
>
> We sincerely appreciate the reviewer's valuable feedback and constructive comments. We have carefully addressed all concerns with detailed responses and would be deeply grateful if the reviewer could kindly review our clarifications before the final decision.

---

> ### Comment · Reviewer_Reek · 2025-11-25
> **Thank authors' rebuttal**
>
> Thank for author's rebuttal. We especially thank authors new experiments for RMSE.
>
> However, we disagree with authors' claim point that outputs exceeding 10k tokens are relatively rare in real-world applications. Nowaday, reasoing model like deepseek and qwen require COT to think over different path of reasoing and using test-time scaling to trade accuracy with context length. We think the reference LONGWRITER from SFT stage can't represent the current stage of reasoing models and their context length and performance.
>
> Thank again for authors' rebuttal, we decide to maintain our current scores.

---

> ### Author Response · Authors · 2025-11-30
> **Long CoT Scenario Evaluation**
>
> We thank the reviewer for the follow-up comments and for acknowledging our additional RMSE experiments. We appreciate the perspective that modern reasoning models—such as DeepSeek and QwQ—often generate very long Chain-of-Thought (CoT) traces, especially under test-time scaling. We agree that CoT lengths in current practice can be substantially long, and this is an important consideration for evaluating prediction robustness.
>
> To address this point more directly, we conducted new experiments specifically targeting the *extreme long-CoT regime*. Using the Floppanacci/QWQ-LongCOT-AIMO dataset [1,2], we filtered for examples with **CoT lengths exceeding 10k tokens**, and compared EGTP against strong baselines on this challenging subset. The results are shown in Table 3.
>
> **Table 3. Performance comparison on Long-CoT (>10k) subset.**
>
> |      | EGTP (Ours)          | SSJF-Reg        | SSJF-MC          | S3               | PiA              | TPV              | TRAIL           | LTR-C            |
> | ---- | -------------------- | --------------- | ---------------- | ---------------- | ---------------- | ---------------- | --------------- | ---------------- |
> | MAE  | **1146.47 ± 147.43** | 2152.50 ± 86.29 | 2718.35 ± 330.13 | 1846.30 ± 405.05 | 2762.66 ± 66.24  | 4066.30 ± 492.20 | 1740.05 ± 77.10 | 2740.62 ± 232.09 |
> | RMSE | **1481.81 ± 233.32** | 2778.71 ± 70.62 | 3776.22 ± 78.44  | 2314.82 ± 511.94 | 3944.59 ± 191.24 | 4838.08 ± 573.73 | 1888.26 ± 58.75 | 3018.25 ± 206.53 |
>
> These results indicate that EGTP remains robust even under the extreme CoT lengths characteristic of strong contemporary reasoning systems. In particular, our method achieves **34.1% lower MAE** compared to the strongest baseline (TRAIL), and substantially outperforms prior predictors such as PiA and TPV, which degrade more significantly in this regime.
>
> We hope this additional experiment directly addresses the reviewer’s concern about applicability to modern long-CoT reasoning models. We sincerely appreciate the reviewer’s time and constructive feedback.
>
> [1] [https://huggingface.co/datasets/Floppanacci/QWQ-LongCOT-AIMO](https://huggingface.co/datasets/Floppanacci/QWQ-LongCOT-AIMO)
> [2] NUMINA: A Natural Understanding Benchmark for Multi-dimensional Intelligence and Numerical Reasoning Abilities

---

### Official Review · Reviewer_bqVo · 2025-11-01

**Soundness:** 3
**Presentation:** 3
**Contribution:** 2
**Rating:** 4
**Confidence:** 4

**Summary:**

This paper proposes two methodologies for regressing the length of a generation from an autoregressive model like an LLM. In EGTP, a small attention-like head is applied over a featurization of the model's activations to regress the length, whereas in PLP a prediction is instead made of the length-to-go at each generation step. The authors compare their predictions against many prior methods, and find that EGTP pooling almost always performs better in accuracy and throughput.

**Strengths:**

1. The problem of length regression is basic and fundamental, yet unsolved. Current methods seem to be very engineered (as opposed to clean and lightweight), and so it appears a good problem to be working on. The authors have made substantial progress on this problem with a fairly simple addition (one prediction head and a couple hyperparameters) that outperforms existing methods, which they demonstrate convincingly. I think overwhelmingly convincing experiments are really needed in these days of LLM research (we have to *know* if something is best in order to build on it!), so I think it's a big strength that the results are so clear and stable across hyperparameters/tasks.

2. The authors also introduce a benchmark with different length distributions than previous ones and a carefully-made soup of various datasets. While sometimes I feel that introducing extra benchmarks can be misleading or contribute to noise, I think in this work it does well to complement LMSYS. This goes back to Strength (1), but I love seeing gains on a diverse set of tasks on datasets with different length distributions made by different people.

**Weaknesses:**

1. Many of the design decisions feel a bit unmotivated (such as binning the regression with exponential tails), and there are many reasonable alternatives that were not mentioned or ablated (such as smoothing, predicting on nonlinear scale, adding small noise to the labels, etc.). It is hard to debate the fact that the authors' methodology works (on many tasks and with different model types and sizes), but it is also very hard to know which parts are necessary and which are suboptimal. If the authors could discuss a bit more about which other methods they considered/tried that didnt work out, and why their solution is a natural one to arrive at?

2. I would have liked the authors to say a bit more about some of the interesting things they saw while training these. In particular, from what I can see in your Appendices D and E it seems like the task doesnt get much easier/harder as model scale changes, which I find interesting. Were there trends like this that you saw, perhaps with respect to model class (Qwen vs Llama vs ...) or optimizer for learning the prediction head or anything? An extra small gain to including such observations is that it would make the paper feel a bit more sciency and less industrial/SOTA-seeking (this is definitely my own bias, but perhaps less jargon could help out; what do you think?).

**Questions:**

1. Aside from the questions in the Weaknesses section, I really like the plot in Figure 2 as a way to explore the role of entropy in this prediction task. I understand that it validates the EGTP method, but I would very much appreciate if the authors could share a little more about what the entropic nature of the problem really is. To me, the success of this entropy-based pooling over other pooling methods suggests that entropy is intimately related with a model's own internal understanding of when it will finish generating. It makes sense for most of the paper to be devoted to presenting this methodology and experiments in the usual SOTA-claiming style, but I would like the authors to discuss a bit more about what is to be learned from this.

2. I think it is important for the authors to say more about how the hidden states were selected. Are the choices the same (or along similar lines) for different model types and sizes, or are they custom for each one? What are some trends that the authors noticed/guidelines for choosing these? I recall in the computer vision days there was great interest in using hidden activations of VGG and ResNet models for prediction of other stuff, and as time went on and models got bigger it became harder to understand the benefits or drawbacks of certain design decisions. I am curious how the analogous story goes in the LLM age, and what your experience with this method has been like.

---

> ### Author Response · Authors · 2025-11-23
>
> > Many of the design decisions feel a bit unmotivated (such as binning the regression with exponential tails), and there are many reasonable alternatives that were not mentioned or ablated (such as smoothing, predicting on nonlinear scale, adding small noise to the labels, etc.). It is hard to debate the fact that the authors' methodology works (on many tasks and with different model types and sizes), but it is also very hard to know which parts are necessary and which are suboptimal. If the authors could discuss a bit more about which other methods they considered/tried that didnt work out, and why their solution is a natural one to arrive at?
>
> For other considered methods, we compared EGTP, pure MSE and pure Cross Entropy as loss function (**As discussed in Appendix D ABLATION STUDY**). MSE fails because sequence lengths exhibit high variance and long tails (Figure 1 in paper), causing training instability. Cross-entropy fails because it treats bins as unordered categories, predicting bin 1 vs. bin 100 has the same penalty as bin 1 vs. bin 2, leading to low loss but high error. While EGTP combines both strengths: binning reduces noise (from classification) while MSE preserves ordinal structure (from regression).
>
>
> **Motivation for Binning:** Our choice of binning the regression targets is motivated by its prevalence and effectiveness in length prediction tasks. Binning is a widely adopted technique that naturally handles the uncertainty in sequence length distributions. This approach has demonstrated success in recent efficient inference literature, such as SSFJ (Interactive LLM Serving with Proxy Model-based Sequence Length Prediction) and S3 (Increasing GPU Utilization during Generative Inference).
>
> For alternatives like Label smoothing, Log-scale Regression and Label Noise, we would like to clarify that they are general-purpose training techniques for improving model performance, rather than methods specifically designed for length prediction tasks. Therefore, they were not discussed in our paper.
> Nevertheless, to address your concerns about different alternatives, we have conducted additional ablation experiments comparing our method and alternatives you mentioned. We use Qwen2.5-7B model in this experiment, and report their MAE. While Log-scale regression shows competitive performance on the Reasoning task, EGTP achieves the best overall performance across all three domains, particularly in LongSeq and RL tasks, demonstrating better robustness.
>
> (Table 1: Comparison with Alternatives)
> | Method | LongSeq | Reasoning | RL |
> | --- | --- | --- | --- |
> | EGTP (Ours) | **83.65±1.28** | 139.14±5.13 | **92.78±2.32** |
> | Label Smoothing | 158.44±0.73 | 144.93±19.60 | 268.88±3.20 |
> | Log-scale Regression| 135.73±2.04 | **136.09±4.76** | 208.02±0.50 |
> | Label Noise | 119.05±2.25 | 217.44±1.81 | 201.77±1.91 |
>
>
>
> > I would have liked the authors to say a bit more about some of the interesting things they saw while training these. In particular, from what I can see in your Appendices D and E it seems like the task doesnt get much easier/harder as model scale changes, which I find interesting. Were there trends like this that you saw, perhaps with respect to model class (Qwen vs Llama vs ...) or optimizer for learning the prediction head or anything? An extra small gain to including such observations is that it would make the paper feel a bit more sciency and less industrial/SOTA-seeking (this is definitely my own bias, but perhaps less jargon could help out; what do you think?).
>
> Regarding your observation on model scale, we found that prediction performance remains relatively stable across different sizes (0.5B to 7B). This aligns with prior work such as SSJF [2], which demonstrated that even small OPT models can achieve reasonable prediction accuracy. We attribute this to the nature of the task: length prediction relies primarily on high-level semantic understanding rather than complex reasoning. Smaller models already capture sufficient semantic cues for this purpose, meaning the advanced reasoning capabilities of larger models provide limited additional benefit for this specific regression task.

---

> ### Author Response · Authors · 2025-11-23
>
> > I think it is important for the authors to say more about how the hidden states were selected. Are the choices the same (or similar lines) for different model types and sizes, or are they custom for each one? What are some trends that the authors noticed/guidelines for choosing these? I recall in the computer vision days there was great interest in using hidden activations of VGG and ResNet models for prediction of other stuff, and as time went on and models got bigger it became harder to understand the benefits or drawbacks of certain design decisions. I am curious how the analogous story goes in the LLM age, and what your experience with this method has been like.
>
> In the era of VGG and ResNet, it was common practice to use the feature maps from the final layer and replace the classification head to adapt models to different datasets or tasks. This paradigm remains prevalent in the large language model era. For instance, in RAG (Retrieval-Augmented Generation) [3] systems, LLMs are used to encode text, and the activations from the final layer are employed for ranking, retrieval, classification, and other downstream tasks. This suggests that the final layer outputs of LLMs contain rich semantic information. Therefore, we follow this established paradigm and utilize the final layer activations for length prediction.
>
> To further validate the rationale of using the final layer, we conducted experiments on the GSM8k dataset using hidden states from different layers of Qwen2.5-0.5B (which has 24 layers) to predict output length. As shown in Table 3, the performance of the final layer (Layer 24) is comparable to that of the best-performing intermediate layers (such as Layer 12 and Layer 16). Layer 24 achieves an RMSE of 92.95 ± 2.80, which is close to Layer 12's 93.47 ± 1.50. Although Layer 12 slightly outperforms in terms of MAE (70.26 ± 1.20), Layer 24's MAE (73.93 ± 2.10) still maintains a competitive level. Considering that using the final layer avoids over-engineering and unnecessary implementation complexity, we consistently utilize the hidden states from the last transformer layer across all model types and sizes in our final implementation.
>
> (Table 2: Performance of using different layers (lower is better ↓))
> |  | RMSE ↓ | MAE ↓ |
> |-------|------|-----|
> | Layer 1 | 110.18 ± 2.50 | 85.91 ± 1.82 |
> | Layer 8 | 97.45 ± 1.81 | 81.29 ± 1.50 |
> | Layer 12 | 93.47 ± 1.50 | **70.26 ± 1.20** |
> | Layer 16 | 94.30 ± 1.64 | 70.79 ± 1.36 |
> | Layer 24 | **92.95 ± 2.80** | 73.93 ± 2.10 |
>
>
> > Aside from the questions in the Weaknesses section, I really like the plot in Figure 2 as a way to explore the role of entropy in this prediction task. I understand that it validates the EGTP method, but I would very much appreciate if the authors could share a little more about what the entropic nature of the problem really is. To me, the success of this entropy-based pooling over other pooling methods suggests that entropy is intimately related with a model's own internal understanding of when it will finish generating. It makes sense for most of the paper to be devoted to presenting this methodology and experiments in the usual SOTA-claiming style, but I would like the authors to discuss a bit more about what is to be learned from this.
>
> To address your interest in the "entropic nature" of the problem, our approach is grounded in recent findings on LLM internal dynamics. Studies such as Step-Entropy [1] suggest that entropy distributions encode the "necessity" and redundancy of future tokens property intrinsically linked to output length. Guided by these insights, we hypothesized that entropy could serve as a robust signal for prediction. Our experimental results validate this connection: we observed that entropy-weighted representations (EGTP) significantly outperform standard pooling, confirming that entropy effectively captures the "information density" required to estimate generation length.
>
> References:
>
>
> [1] Li, Y., et al. "Compressing Chain-of-Thought in LLMs via Step Entropy."
>
> [2] Yuksekgonul, M., et al. "SSJF: Interactive LLM Serving with Proxy Model-based Sequence Length Prediction."
>
> [3] Li, Y., et al. "Towards Agentic RAG with Deep Reasoning: A Survey of RAG-Reasoning Systems in LLMs." 2025.

---

> ### Author Response · Authors · 2025-11-23
>
> We thank the reviewer for the constructive feedback. We have carefully addressed the raised concerns and hope this clarification may support a more favorable reassessment.

---

> ### Author Response · Authors · 2025-11-24
>
> We sincerely appreciate the reviewer's valuable feedback and constructive comments. We have carefully addressed all concerns with detailed responses and would be deeply grateful if the reviewer could kindly review our clarifications before the final decision.

---

> ### Author Response · Authors · 2025-11-27
>
> We appreciate the reviewer’s constructive comments. We have carefully addressed all concerns. We hope these clarifications will facilitate a favorable re-evaluation.

---

> > ### Comment · Reviewer_bqVo · 2025-11-28
> >
> > I thank the authors for their clear responses and supporting experimental results. My questions regarding binning and the use of final-layer activations were answered comprehensively and convincingly. Regarding entropy, I see now that this intuition (connection between entropy, a measurement of the model's "generation-to-go", and length prediction) is generally present in the broader length prediction literature, and this paper's investigation of EGTP builds on this intuition and makes it concrete. Given these developments, I would like to raise my score from a 4 to a 6 (though OpenReview is perplexingly not allowing me to edit the scores right now).

---

### Official Review · Reviewer_J52D · 2025-11-01

**Soundness:** 3
**Presentation:** 3
**Contribution:** 3
**Rating:** 6
**Confidence:** 2

**Summary:**

The paper addresses the problem of estimating the output length of LLMs, which is an important problem for efficient batching during inference. If LLM generations vary a lot in length, batching will naturally waste compute because of the introduced padding.

The proposed method directly reuses the LLM’s internal activations instead of relying on separate auxiliary predictors. It applies two key methods: 1) Entropy-guided token pooling, 2) Progressive length prediction to handle stochastic generation.

**Strengths:**

- The paper is written clearly and easy to follow.
- The problem is well motivated.
- I am not too familiar with the topic, but after a short literature research I tend to agree that a benchmark as proposed in the paper is a reasonable contribution to the community.

**Weaknesses:**

- Although a main selling point for the method is efficiency, I could not find any cost comparison to methods that use an auxiliary model for prediction.

**Questions:**

I understood: The proposed length regression relies on pooled hidden representations extracted by the LLM we want to perform inference on. The length information can be used for efficient batching of the prompts, such that minimal padding will occur. What I do not understand is: To get the representations, we already need to process the inputs (possibly batched?) by the LLM. Did I miss something?

---

> ### Author Response · Authors · 2025-11-23
>
> > Although a main selling point for the method is efficiency, I could not find any cost comparison to methods that use an auxiliary model for prediction.
>
> We have provided a cost comparison in Section 4.4 (End-to-End System Performance) to demonstrate our method's efficiency. To offer a more granular quantitative analysis, we evaluated the overhead of various auxiliary models using the GSM8K math reasoning dataset. For each model, we measured the inference latency and memory consumption required to predict response length. Results are presented in Tables 1 & 2. "Avg Time" represents the average wall-clock time for the auxiliary model to process a query and generate a prediction. All experiments were conducted on a single NVIDIA RTX 4090 (24GB) GPU under consistent environmental settings.
>
> Note: Unlike other baselines, PiA relies on the LLM itself to self-predict the token count rather than using an external auxiliary model. For a fair comparison, we only recorded the time consumed for generating the specific length-prediction tokens.
>
> ### Table 1: Auxiliary Model Overhead for Predicting Qwen2.5-7B Response Lengths
>
> | Metric | EGTP (Ours) | SSFJ-MC | SSFJ-Reg | S3 | LTR-C | TPV | PiA | TRAIL |
> | :--- | :--- | :--- | :--- | :--- | :--- | :--- | :--- | :--- |
> | **Avg Time (ms)** | 0.67+-0.07 | 3.94 +- 0.01 | 4.04 ± 0.12 | 2.26 +- 0.08 | 12.57 +- 0.07 | 0.83 +- 0.21 | 60.90 +- 0.33 | 2.04 +- 0.08 |
> | **Avg. VRAM (MB)** | 7.21 | 269.76 | 269.76 | 264.19 |  238.41 | 238.41 | - | 268.03 |
>
> ### Table 2: Auxiliary Model Overhead for Predicting Llama3.2-3B Response Lengths
>
> | Metric | EGTP (Ours) | SSFJ-MC | SSFJ-Reg | S3 | LTR-C | TPV | PiA | TRAIL |
> | :--- | :--- | :--- | :--- | :--- | :--- | :--- | :--- | :--- |
> | **Avg Time (ms)** | 0.65 +- 0.01 | 3.95 +- 0.02 | 4.18 ± 0.13 | 2.41 +- 0.04 | 12.56 +- 0.03 | 0.79 +- 0.11 | 60.40 +- 1.97 | 2.59 +- 0.11 |
> | **Avg. VRAM (MB)** | 5.52 | 269.76 | 269.76 | 264.19 |  238.41 | 238.41 | - | 268.03 |
>
> EGTP achieves the lowest inference time (~0.66ms) and minimal memory footprint (5.52-7.21 MB) compared to baselines. Model-based methods like SSFJ and TRAIL require 3-4ms with 238-270 MB VRAM, while PiA is slowest at ~60ms due to LLM-based prediction. The results demonstrate that EGTP's entropy-based approach provides a lightweight and efficient solution with negligible computational overhead.
>
>
> > I understood: The proposed length regression relies on pooled hidden representations extracted by the LLM we want to perform inference on. The length information can be used for efficient batching of the prompts, such that minimal padding will occur. What I do not understand is: To get the representations, we already need to process the inputs (possibly batched?) by the LLM. Did I miss something?
>
> In modern LLM serving systems, micro-batching is commonly employed (Zheng et al., 2023). The system receives a batch of prompts and encodes them. We obtain the hidden states in this stage and predict the response lengths. The encoded prompts are then divided into micro-batches to generate responses. We utilize the predicted response lengths to schedule the micro-batch generation process to avoid unnecessary padding.
>
> Reference: Zheng, Z., Ren, X., Xue, F., Luo, Y., Jiang, X., & You, Y. (2023). Response Length Perception and Sequence Scheduling: An LLM-Empowered LLM Inference Pipeline. arXiv preprint arXiv:2305.13144.

---

> > ### Comment · Reviewer_J52D · 2025-11-25
> > **computational cost**
> >
> > Thank you for your additional explanations and experiments, I appreciate your effort.
> >
> > From your additional experiments I conclude that most of the SOTA methods are clearly performing worse compared to your proposed method. There seems to be just one competitor, which is TPV. The speed improvement compared to TPV are minor but present.
> >
> > The reported VRAM consumption of TPV is considerably higher. However, I am not sure about the experimental setup and to not trust that the comparison is fair (i.e. is memory consumed by the base model counted in?). The original paper proposed to only use a two layer regressor that consumed the input embeddings. The reported 238.41MB for this regressor feel extremely large.
> >
> > Considering the explanations and additional experimental results, I decided to keep my score.

---

> > > ### Author Response · Authors · 2025-11-25
> > >
> > > We sincerely apologize for the clerical error regarding TPV's memory consumption. Upon re-checking our logs, the correct auxiliary memory usage for TPV is approximately 6.38 MB (Table 1) and 5.95 MB (Table 2).
> > >
> > > While TPV is as lightweight as our method, we must emphasize that TPV's performance is severely limited (much worse than our method). As demonstrated in Table 1 of our original paper, TPV yields a significantly higher MAE across all scenarios compared to our proposed method. We wish to highlight that performance is clearly a more important metric than memory usage; therefore, we consider TPV to be not competitive.

---

> > > > ### Comment · Reviewer_J52D · 2025-11-26
> > > >
> > > > Thank you for double checking and the clarification. I will contemplate about out discussion and get back to you.

---

> ### Author Response · Authors · 2025-11-23
>
> We thank the reviewer for the constructive feedback. We have carefully addressed the raised concerns and hope this clarification may support a more favorable reassessment.

---

> ### Author Response · Authors · 2025-11-24
>
> We sincerely appreciate the reviewer's valuable feedback and constructive comments. We have carefully addressed all concerns with detailed responses and would be deeply grateful if the reviewer could kindly review our clarifications before the final decision.

---

### Author Response · Authors · 2025-11-30
**Summary of Rebuttal Updates and Reviewer Consensus**

Due to the recent suspension of reviewer-author interactions, we are providing this brief summary to clarify the final status of our reviews and highlight that all raised concerns have been effectively resolved.

1. Consensus on Improvement (Reviewer bqVo, Initial Score: 4, but **promised to raise to 6**)

Reviewer bqVo initially raised a comprehensive set of concerns regarding design alternatives (such as binning versus smoothing), the stability of predictions across model scales, the rationale for hidden state selection, and the theoretical role of entropy. In our rebuttal, we addressed these thoroughly by providing the requested ablation studies, conducting layer-wise analysis, and offering theoretical grounding for the entropy-based approach. The reviewer explicitly acknowledged that these questions were "answered comprehensively and convincingly." **Crucially, they stated their clear intention to raise the score from 4 to 6, noting that the system lock prevented the official update**. We kindly ask the AC to factor this confirmed improvement into the final assessment.

2. Clarification on Pipeline & Baselines (**Reviewer J52D Score: 6**)

Reviewer J52D raised questions regarding the inference pipeline and the memory overhead of auxiliary models. We addressed the pipeline concern by detailing our integration with micro-batching. We also corrected a clerical error in the TPV baseline's memory usage (updating ~238MB to ~6MB), while emphasizing that our method maintains a decisive accuracy advantage regardless of this correction. The reviewer accepted these explanations and confirmed the competitiveness of our approach.

3. Addressing Reviewer Reek’s  (**Reviewer Reek Score: 6**) Concerns: Statistical Rigor & Long-CoT Reasoning

Reviewer Reek requested additional statistical metrics (such as RMSE) and clarification on the distinction from prior work like TRAIL. We successfully addressed these by providing the requested metrics—where our method demonstrated superior stability—and clarifying our entropy-guided mechanism. While satisfied with these responses, the reviewer raised a final valid point regarding the method's applicability to modern Long-Chain-of-Thought (CoT) reasoning models (>10k tokens). To address this, we immediately conducted an additional evaluation on the Floppanacci/QWQ-LongCOT-AIMO benchmark. The results confirm that EGTP retains its superior performance even in this complex setting, effectively dispelling the concern that our success was limited to standard SFT distributions.

With Reviewer bqVo's intended score increase, the clarification of Reviewer J52D's technical query, and the new Long-CoT evidence addressing Reviewer Reek's final point, we believe our work is now robustly supported by all reviewers.

Best regards,

The Authors

---

### Meta-Review · Area_Chair_Y4mi · 2026-01-13

**Summary:**

Across reviewers, the paper was generally regarded as well-motivated, technically sound, and practically relevant, with particular strengths in leveraging internal LLM activations for efficient length prediction and in introducing the ForeLen benchmark. The main concerns focused on:

- Motivation and justification of design choices (e.g., binning strategy, entropy-guided pooling, and hidden-state selection),

- Comparative efficiency and fairness of baselines, especially regarding auxiliary-model overhead,

- Statistical rigor of evaluation metrics, beyond MAE, and

- Generality to modern long Chain-of-Thought (CoT) and very long (>10k token) generation regimes, as well as the degree of novelty relative to prior work such as TRAIL.

The rebuttal and follow-up experiments substantially strengthened the paper by addressing these points with additional ablations, corrected measurements, new metrics, and new long-CoT evaluations. Overall, the reviewer discussion converged toward a positive assessment, with no remaining critical flaws that would preclude acceptance.

**Reviewer Concerns:**

Addressed Concerns

- Design choices and alternatives (Reviewer bqVo):
Thoroughly addressed via new ablations (loss functions, binning alternatives, log-scale regression, label smoothing/noise), layer-wise analyses, and conceptual clarification of entropy’s role. The reviewer explicitly confirmed these were answered “comprehensively and convincingly.”

- Hidden state selection (Reviewer bqVo):
Addressed through empirical evidence showing final-layer representations are competitive with intermediate layers and by articulating a clear, principled guideline aligned with common LLM practice.

- Efficiency and auxiliary-model overhead (Reviewer J52D):
Addressed with explicit end-to-end cost comparisons, clarification of the inference pipeline (micro-batching), and correction of a clerical error in TPV’s memory usage.

- Statistical rigor (Reviewer Reek):
Addressed by adding RMSE and variance analyses, demonstrating EGTP’s stability across tasks.

- Generality to long and super-long sequences (Reviewer Reek):
Addressed via new experiments on super-long datasets (>17k tokens) and an additional targeted evaluation on extreme long-CoT (>10k tokens), showing EGTP remains robust and outperforms baselines.

Remaining / Partially Outstanding Concerns

- Incremental nature relative to prior work (Reviewer Reek):
While clarified (single-stage, entropy-guided use of LLM activations vs. TRAIL’s two-stage approach), the contribution may still be viewed as incremental by some readers, though reviewers did not treat this as a blocking issue.

- Prevalence of very long CoT in real-world deployments (Reviewer Reek):
The authors’ initial framing was challenged, but the subsequent long-CoT experiments substantially mitigated this concern. The reviewer ultimately maintained, rather than lowered, their score.

Overall, no substantive technical concerns remain unresolved.

**Reviewer Scores:**

Reviewer bqVo:
Initial score: 4 → Expected score: 6
The reviewer explicitly stated an intention to raise the score after the rebuttal, citing comprehensive and convincing responses.

Reviewer J52D:
Initial score: 6 → Expected score: 6
Concerns about efficiency and memory were clarified; the reviewer indicated satisfaction and chose to keep the score.

Reviewer Reek:
Initial score: 6 → Expected score: 6
While some reservations about framing and scope remained, the added RMSE analysis and long-CoT experiments addressed the core technical concerns, leading the reviewer to maintain their score.

---

### Decision · Program_Chairs · 2026-01-26

Accept (Poster)